# Neutralization of the *Plasmodium*-encoded MIF ortholog confers protective immunity against malaria infection

Alvaro Baeza Garcia[1,2,3], Edwin Siu[1,2,3], Tiffany Sun[1,2,3], Valerie Exler[1,2,3,4], Luis Brito[5], Armin Hekele[5], Gib Otten[5], Kevin Augustijn[6], Chris J. Janse[6], Jeffrey B. Ulmer[5,9], Jürgen Bernhagen[4], Erol Fikrig [7,8], Andrew Geall[5] & Richard Bucala[1,2,3]

*Plasmodium* species produce an ortholog of the cytokine macrophage migration inhibitory factor, PMIF, which modulates the host inflammatory response to malaria. Using a novel RNA replicon-based vaccine, we show the impact of PMIF immunoneutralization on the host response and observed improved control of liver and blood-stage *Plasmodium* infection, and complete protection from re-infection. Vaccination against PMIF delayed blood-stage patency after sporozoite infection, reduced the expression of the Th1-associated inflammatory markers TNF-α, IL-12, and IFN-γ during blood-stage infection, augmented Tfh cell and germinal center responses, increased anti-*Plasmodium* antibody titers, and enhanced the differentiation of antigen-experienced memory CD4 T cells and liver-resident CD8 T cells. Protection from re-infection was recapitulated by the adoptive transfer of CD8 or CD4 T cells from PMIF RNA immunized hosts. Parasite MIF inhibition may be a useful approach to promote immunity to *Plasmodium* and potentially other parasite genera that produce MIF orthologous proteins.

[1] Department of Internal Medicine, Yale School of Medicine, New Haven, CT 06520, USA. [2] Department of Pathology, Yale School of Medicine, New Haven, CT 06520, USA. [3] Department of Epidemiology of Microbial Diseases, Yale School of Public Health, New Haven, CT 06520, USA. [4] Institut für Schlaganfall- und Demenzforschung, Klinikum der Universität München Ludwig-Maximilians-Universität München, D-81377 München, Germany. [5] Novartis Vaccines, Inc., 350 Massachusetts Avenue, Cambridge, MA 02139, USA. [6] Leiden Malaria Research Group, Department of Parasitology, Leiden University Medical Centre, 2300 RC Leiden, The Netherlands. [7] Section of Infectious Diseases, Department of Internal Medicine, Yale University School of Medicine, New Haven 06520 CT, USA. [8] Howard Hughes Medical Institute, Chevy Chase 20815 MD, USA. [9] Present address: Slaoui Center for Vaccines Research, GSK Vaccines, 14200 Shady Grove Rd., Rockville MD 20850, USA. Correspondence and requests for materials should be addressed to R.B. (email: richard.bucala@yale.edu)

I n 2013, there were approximately 200 million clinical cases and 584,000 deaths from malaria caused by parasites of the genus *Plasmodium*[1]. *Plasmodium* sporozoites enter the skin through the bite of infected *Anopheles* mosquitoes, transit to the liver, and replicate over several days to produce merozoites, which then initiate an erythrocytic cycle of infection that produces the clinical manifestations of malaria[2]. Immunologically naïve hosts are at the highest risk of lethal malaria but survivors may develop partial immunity and tolerance to disease manifestations. Such partial protection does not prevent re-infection and declines in the absence of re-exposure to parasites[2,3]. One mechanism for failure to develop sterilize immunity may be the inability of the infected host to achieve immunologic memory and maintain an effective anti-parasite immune response[4,5].

The cellular processes responsible for ineffective immunity to malaria are unclear, although studies support an impaired development of the adaptive response with poor establishment of germinal centers (GC) and a disruption of their architecture in the spleen[6–8]. Effective GC formation requires CD4 T follicular helper (Tfh) cells, which may be downregulated by an unresolved pro-inflammatory response and the expression of TNF-α, IL-12, IFN-γ, and T-bet[9,10]. How *Plasmodium* infection negatively impacts GCs is not understood, although parasite factors are likely to play a central role[2,4].

Many parasitic pathogens, including all studied *Plasmodium* species, express an ortholog of the mammalian cytokine macrophage migration inhibitory factor (MIF)[11,12]. In studies of the erythrocytic stage of *Plasmodium berghei* ANKA (*Pb*A) malaria, *Plasmodium* MIF (PMIF) was observed to be secreted into infected erythrocytes and released upon schizont rupture[13]. PMIF elicits a MIF receptor-dependent inflammatory response that interferes with the differentiation of *Plasmodium*-specific CD4 T effector cells into long-lived memory precursors by increasing the expression of TNF-α, IL-12, IFN-γ, and T-bet[14]. The role of PMIF also has been examined in the *Plasmodium* liver-stage of infection. Genetically-targeted *Plasmodium* strains that lack PMIF do not show defects in virulence or in life cycle, however infection with PMIF-deficient *Plasmodium yoelii* may be associated with retardation of parasite growth in liver and a delay in blood-stage patency[15].

Given the potential role of PMIF in modulating the immune response and in liver-stage parasite development, we investigated herein the impact of genetic deletion or immunoneutralization of PMIF in the *Pb*A experimental model of severe malaria. BALB/c mice infected with *Pb*A*mif*− blood-stage parasites showed a robust induction of antibody-secreting plasma cells and improved differentiation of germinal Tfh cells when compared to wild type parasites. PMIF immunization in turn recapitulated the phenotype observed with the *Pb*A*mif*− parasites, with improved development of CD4 T effector cells into long-lived memory precursors and enhanced differentiation of Tfh cells and antibody-secreting B cells. PMIF-immunized mice showed improved control of liver-stage infection that was associated with an increase in the number *Plasmodium*-specific liver-resident memory CD8 T cells. PMIF immunization also enhanced host control of the first infection and conferred complete protection to re-infection.

## Results

**PMIF impairs germinal center formation during malaria.** Human and experimental mouse studies suggest that strong pro-inflammatory responses generated during blood-stage infection can inhibit productive GC and Tfh cell responses[7,8], and recent data suggest a role for PMIF in the suppression of CD4 T cell differentiation[14]. To assess whether PMIF pro-inflammatory

activity affects the GC responses, we infected BALB/cJ mice with *Pb*AWT or *Pb*A*mif*− parasites. Infection with both strains results in equivalent parasitemia and splenic parasite burden, and comparable levels of circulating host MIF[14]. The frequency and total numbers of GC B cells (CD19+CD38loGL7+) in the spleens of *Pb*A*mif*− infected mice was significantly increased when compared to *Pb*AWT-infected mice (Fig. 1a and Supplementary Figure 1a). The frequency and number of memory B cells (CD19+IgD−CD138−CD38hi) also was significantly lower in the *Pb*AWT-infected mice than in the mice infected with *Pb*A*mif*− parasites (Fig. 1b), and this was associated with a 5-fold increase in the parasite-specific antibody response (Fig. 1c). Immunohistochemical staining at 15 days after infection of spleen sections from *Pb*AWT mice showed a significant loss of the T cell zone and a disorganized follicular architecture when compared with *Pb*A*mif*− infected mice. Taken together, these data suggest that PMIF impairs GC reactions and antibody responses during experimental malaria infection.

**PMIF decreases Tfh cell responses during malaria infection.** Tfh cells are essential for the formation and maintenance of GCs and enable proper B cell development into antibody-producing plasma cells and memory B cells[16]. We investigated if the impairment in GC formation associated with PMIF was a consequence of defective Tfh differentiation. We examined the frequency and number of activated Tfh cells (CD4+CD62L−CXCR5hiPD-1hi) in the spleens of mice at days 6 and 15 after infection with *Pb*AWT or *Pb*A*mif*− parasites. Mice infected with *Pb*A*mif*− parasites showed a significant increase in the number of Tfh activated cells at day 6 when compared to *Pb*AWT-infected mice (Fig. 2a, b and Supplementary Figure 2a). This difference was maintained at day 15 of infection, and without a change in the number of measured Tfh cells in the *Pb*AWT-infected mice. We also investigated if the difference in the number of Tfh cells between the two groups was due to a defect in their maturation[7], despite a similar percentage of pre-Tfh cells in mice infected with *Pb*AWT or *Pb*A*mif*−. The number of pre-Tfh cells (CD4+CD62L−CXCR5intPD-1int) was significantly elevated after 6 days of infection with *Pb*AWT (Fig. 2c). We measured the expression of the transcription factor Bcl-6, a regulator of Tfh cell differentiation[17]. Bcl-6 expression was higher in the *Pb*A*mif*- infected mice, suggesting a defect in the maturation of these cells in the presence of PMIF (Fig. 2d).

In the setting of malaria infection, the balance between Th1 and Tfh responses is determined by the expression of T-bet and Bcl-6. Excessive expression of T-bet represses Bcl-6 expression and interferes with Tfh cell expansion and GC formation[7,9]. In the presence of PMIF, there is an evidence of increased expression of the transcription factor T-bet by *Plasmodium*-responsive CD4 T cells as a consequence of elevated host production of IL-12 and IFN-γ[14]. We investigated the effect of PMIF in driving Tfh responses toward Th1 development by measuring T-bet expression in the Tfh lineage cells. T-bet was significantly higher in the CD4+CD62L−Bcl6hi splenic cells of mice infected with *Pb*AWT than *Pb*A*mif*− parasites (Fig. 2e). In accordance with this increase in T-bet expression, there also was an elevation in the expression number of both pre-Tfh and Tfh cells in mice infected with *Pb*AWT when compared to those infected with *Pb*A*mif*− (Supplementary Figure 2b, c). Finally, Tfh cells from *Pb*AWT-infected mice expressed higher levels of the cytokine IFN-γ (Supplementary Figure 2d). These data suggest that a Th1 pro-inflammatory response driven by PMIF during acute infection has a detrimental effect on the development of responsive Tfh cells, leading to inadequate GC formation.

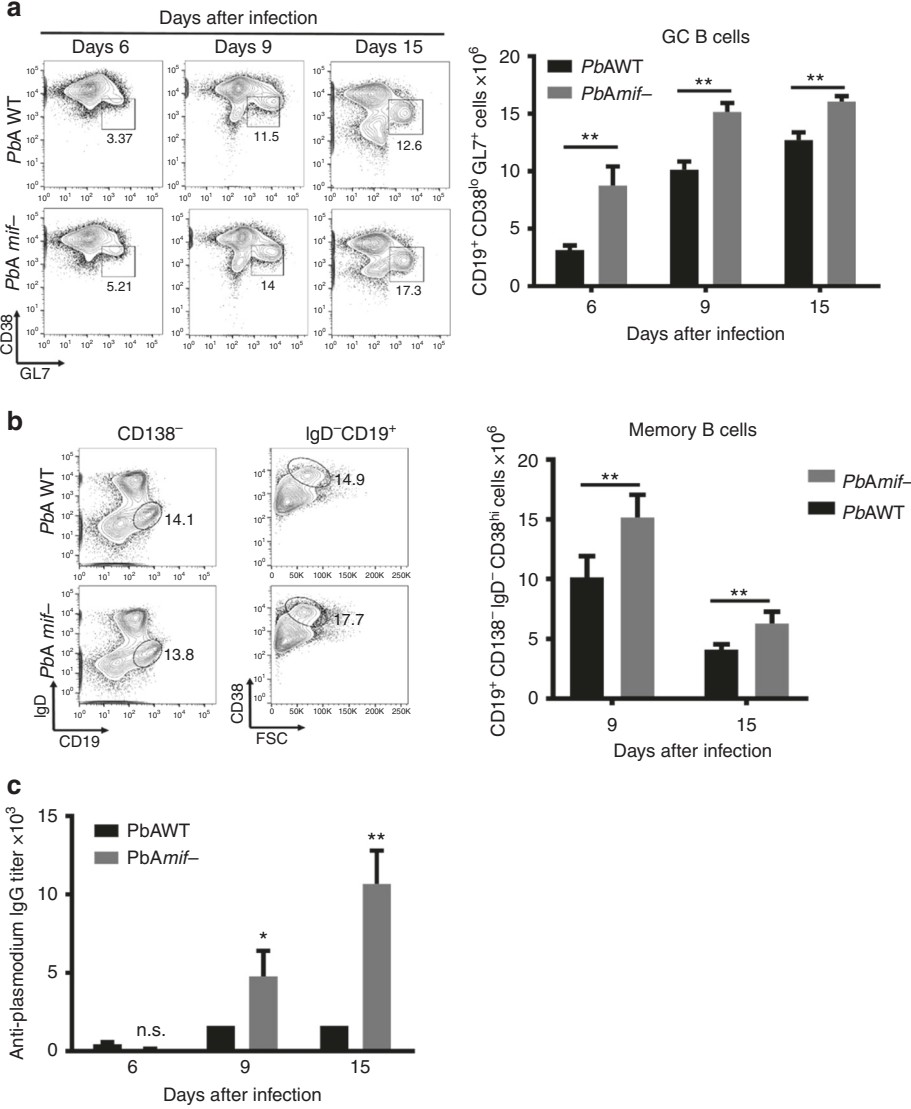

**Fig. 1** PMIF impairs germinal center formation. BALB/cJ mice were infected with $10^6$ *Pb*AWT or *Pb*A*mif*− iRBCs. On day 6, 9, and 15, splenocytes were isolated and the total number of **a** germinal center (CD19+CD38loGL7+) and **b** (CD19+CD138−IgD−CD38hi) memory B cells were determined. Results are from three separate experiments. Bars represent the mean of 12 mice ± SD. **$p < 0.05$ by Mann–Whitney test. **c** Anti-*Plasmodium* antibodies titers from BALB/cJ mice that were infected with *Pb*AWT or *Pb*A*mif*− iRBCs. On day 6, 9, and 15, sera of infected mice were collected, and the *Plasmodium*-specific IgG responses measured by ELISA. Results are from three separate experiments. Bars represent the mean of 12 mice ± SD; n.s.: $p > 0.05$; *$p < 0.05$; **$p < 0.01$ by two-way ANOVA

**PMIF influences PbA development in liver.** Blood-stage infection with *Pb*A*mif*− parasites results in an augmented memory CD4 T cell response when compared to infection with *Pb*AWT, although the survival of infected hosts is unchanged[14]. To examine the impact of PMIF on liver-stage parasite development, which may be impaired in the *P. yoelii* model[15], BALB/cJ mice were infected with 2000 freshly isolated *Pb*AWT or *Pb*A*mif*− sporozoites and blood-stage patency assessed. All mice infected with *Pb*AWT sporozoites developed patent infection at 3 days; by contrast, fewer than 10% of mice infected with *Pb*A*mif*− parasites showed blood-stage patency at 5 days and 25% of mice remained free of parasitemia at 21 days (Supplementary Figure 3a). The livers and spleens of mice infected with *Pb*AWT or *Pb*A*mif*− sporozoites were harvested 7 days after infection and the CSP (Circumsporozoite protein) epitope-specific CD8 T cell responses assessed by flow cytometry. CSP-specific CD8 T cells (CD8+CD11ahiTetrCSPhi) were identified in both groups of mice but increased numbers were evident in the livers of mice infected with *Pb*A*mif*− (Supplementary Figure 3b). The phenotype of CSP-specific CD8 T cells was further characterized by the expression of CD44, CD69, and KLGR1 to better differentiate between the main subsets of CSP-specific cells. We found two distinct populations of CD8 T cells in the livers of *Pb*AWT and *Pb*A*mif*− mice: resident memory cells (Trm: CD44hiKLGR1−CD69+) and effector memory cells (Tem: CD44hiKLGR1hiCD69−), which are two populations described recently to persist long term and to be essential for protection against re-infection[11]. Consistent with the CSP-specific cell results, the number of liver Trm and Tem cells was significantly lower in the *Pb*AWT than the *Pb*A*mif*− infected mice (Supplementary Figure 3c), with Trm cells representing ~64% of the total intrahepatic CSP-specific cells. These data support the notion that PMIF deficiency impairs liver-stage parasite development, as previously suggested[15], and this impairment is associated with an augmentation in the liver-resident CD8 T cell response.

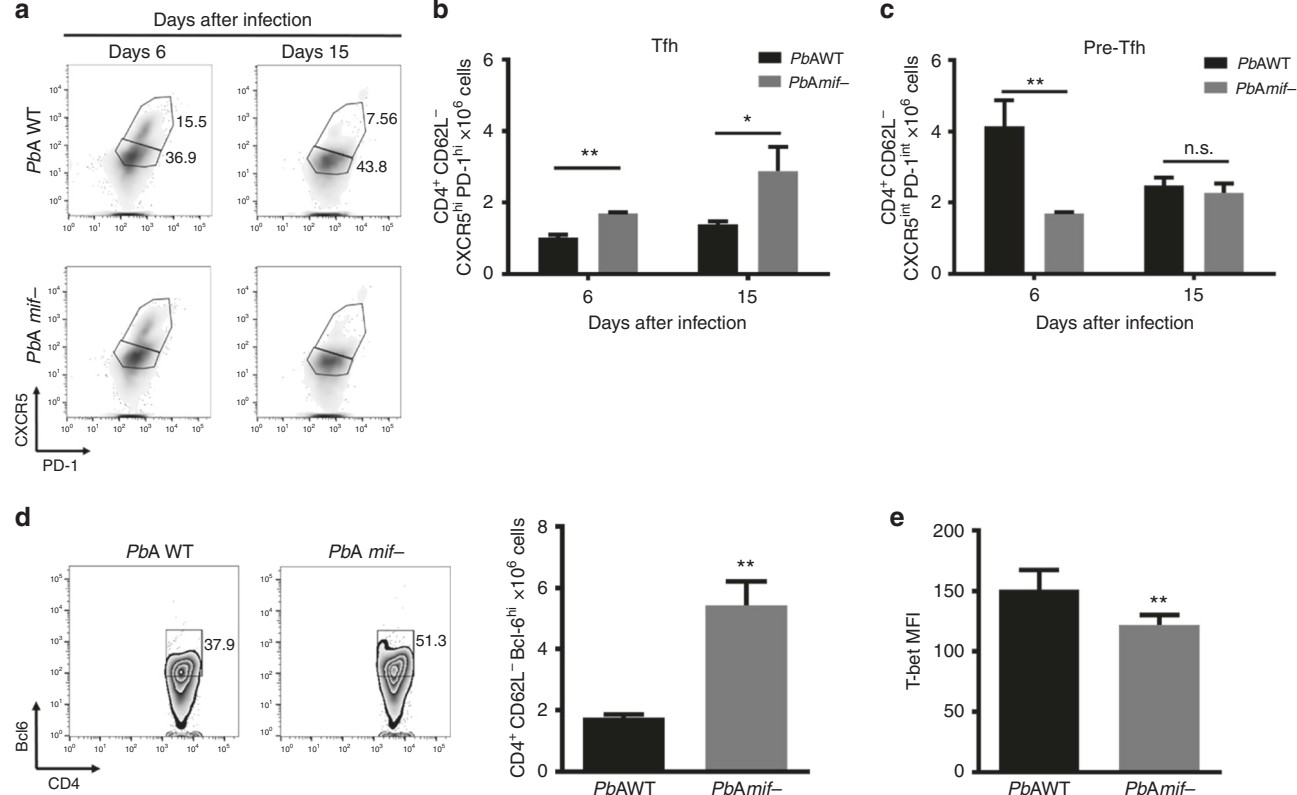

**Fig. 2** PMIF inhibits Tfh cell development. BALB/cJ mice were infected with $10^6$ PbAWT or PbAmif− iRBCs. On days 6 and 15 after infection, splenocytes were isolated and Tfh cells assessed. **a–c** Representative plots and absolute numbers of Tfh and pre-Tfh cells in the two groups of mice. Results are from three separate experiments. Bars represent the mean of 12 mice ± SD (*$p < 0.05$, **$p < 0.001$ by Mann–Whitney test). **d** Representative plots and absolute number of CD4+CD62L−Bcl-6hi cells at day 6 after infection, and **e** expression of transcription factor T-bet in CD4+Bcl-6hi cells. Results are from three separate experiments. Bars represent the mean of 12 mice ± SD; *$p < 0.05$, **$p < 0.001$ by Mann–Whitney test

**An RNA replicon encoding PMIF elicits humoral and cellular immunity**. The present and previous[14] observations suggest immunoregulatory actions for PMIF in the development of anti-Plasmodium CD4 central memory T cells, GC Tfh responses, and B cell maturation. We hypothesized that inhibition of PMIF activity could improve host immunity against Plasmodium infection and potentially confer protection against re-infection. For immunization of naïve mice, we subcloned pmif into a self-amplifying mRNA "replicon", which is an antigen delivery methodology that elicits cellular and humoral responses without generating a limiting anti-vector response[18,19]. We studied the impact of pmif or control RNA immunization on liver- and blood-stage PbAWT infection in BALB/cJ mice, which under normal circumstances results in a progressive parasitemia and death from anemia at 2–3 weeks. If cured by chloroquine treatment, the initially infected mice remain susceptible to the second infection and develop a patent parasitemia that persists for at least 10 days[20,21].

We assessed the immunogenicity of PMIF in mice given two sequential pmif RNA replicon immunizations followed by challenge infection with PbAWT-infected red blood cells (iRBCs) (Supplementary Figure 4a). Single immunization resulted in a primary anti-PMIF antibody response that increased in titer by 4-fold after the second immunization (Supplementary Figure 4a). Anti-PMIF antibody development was associated with the elicitation of PMIF specific CD4 T cells (Supplementary Figure 4c), and the elicited anti-PMIF IgG neutralized the stimulatory action of PbAWT iRBCs or recombinant PMIF on inflammatory cytokine production by bone marrow-derived macrophages (Supplementary Figure 4d, e). As host MIF

deficiency may alter the course of Plasmodium infection[22], we also tested the specificity of the antibody response in the PMIF-immunized mice and found that anti-PMIF IgG from immune serum neutralized PMIF upregulation of host TLR4 expression but failed to detect mouse MIF (Supplementary Figure 4f–h). To exclude a potential contribution for elicited anti-PMIF in the clearance of free parasites after schizont release, we also confirmed that PMIF is expressed only in the cytosolic fraction and not on membranes (Supplementary Figure 4i). As a specificity control, we studied the impact of anti-PMIF IgG in mice infected with PbAmif− iRBCs, which show similar parasitemia and lethality in this model as mice infected with PbAWT[14]. Anti-PMIF IgG administration to mice infected with PbAmif− did not influence parasitemia, and disease course resembled that of PbAWT-infected mice treated with a non-immune (Con) IgG (Supplementary Figure 4j). These results demonstrate that pmif RNA replicon immunization elicits both a cellular and humoral immune response against PMIF, and that anti-PMIF IgG blocks the pro-inflammatory action of PMIF without inhibiting the action of host MIF.

**An RNA replicon encoding PMIF confers protection to re-infection**. Mice immunized with pmif or control RNA replicons were injected with $10^6$ PbAWT-infected iRBCs and the progress of infection followed over time. There was a more rapid increase in parasitemia after day 5 in the control group, which became moribund on day 21 (Fig. 3a, b). By contrast, the pmif RNA replicon immunized mice showed better control of parasitemia during the first 15 days of infection and a 37% prolongation in

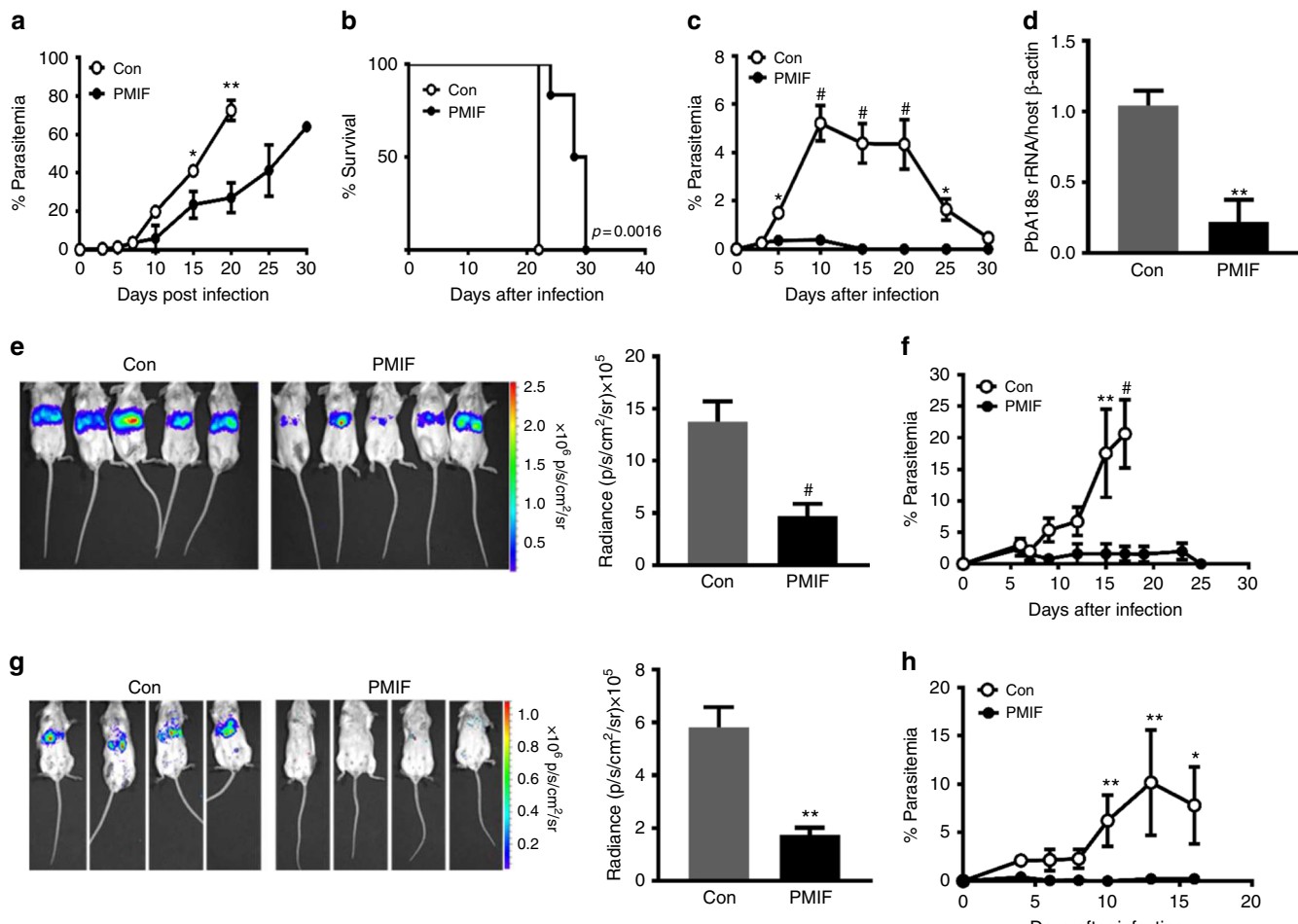

**Fig. 3** PMIF neutralization confers complete protection to re-infection by wild type *P. berghei* ANKA. **a** Parasitemia after infection of BALB/cJ mice ($10^6$ *Pb*AWT iRBCs) previously immunized with RNA replicons encoding PMIF (black circle) or a control (Con) RNA (white circle); *$p < 0.05$, **$p < 0.01$, by two-way ANOVA and error bars denote ±SD. **b** Kaplan–Meier survival plots for immunized mice following infection with *Pb*AWT (black circle, PMIF and white circle, Con). Data are from two independent experiments with 10–15 animals per group; **$p = 0.0016$ by log-rank (Mantel Cox) test. **c** Percentage of iRBCs in BALB/cJ mice previously immunized with RNA encoding PMIF (black circle) or Con RNA (white circle), treated with chloroquine, and re-infected with $10^6$ *Pb*AWT iRBCs; *$p < 0.05$, #$p < 0.0001$ by two-way ANOVA and error bars denote ±SD. **d** Splenic parasite load 6 days after reinfection with iRBCs was measured by quantitative PCR of *Pb*AWT 18S rRNA relative to host β-actin. Results are from two separate experiments. Bars represent the mean of 6 mice ± SD; **$p < 0.01$ by Mann–Whitney test. **e** *Pb*Aluc liver load and absolute luminescence values in PMIF (black circle) or Con (white circle) RNA replicon immunized mice 48 h after the first infection with 2000 *Pb*Aluc sporozoites. **f** Percentage of iRBCs after the first infection of BALB/cJ mice with 2000 *Pb*Aluc sporozoites. Data are from two independent experiments. Bars represent the mean of 10 mice ± SD; **$p < 0.01$, #$p < 0.0001$ by Mann–Whitney and two-way ANOVA. **g** *Pb*A liver load and absolute luminescence values in PMIF (black circle) or Con (white circle) RNA replicon immunized hosts 48 h after the second infection with 2000 *Pb*Aluc sporozoites. **h** Percentage of iRBCs after the second infection of BALB/cJ mice. Data are from two independent experiments. Bars represent the mean of 10 mice ± SD; *$p < 0.05$, **$p < 0.01$ by Mann–Whitney test and two-way ANOVA

mean survival time. To test for the development of a protective memory response, a cohort of *pmif* RNA replicon immunized and *Pb*AWT-infected mice was cured by treatment with chloroquine and re-infected 4 weeks later (see scheme Supplementary Figure 4a). Mice that received the control RNA replicon developed a rapidly increasing parasitemia that was resolved at day 30. By contrast, patent parasitemia did not develop in the *pmif* RNA replicon immunized mice, nor were parasites detected in organs (Fig. 3c, d). Challenge infection was associated with a further increase in anti-PMIF titer, indicating that PMIF immunization produces a humoral response that persists after blood-stage infection and is rapidly activated after the second challenge (see Supplementary Figure 4b, third titer).

We studied the effect of PMIF on pre-erythrocytic stage *Plasmodium* by immunizing mice with *pmif* or control RNA replicons followed by i.v. injection of 2000 *Pb*A expressing luciferase (*Pb*Aluc) sporozoites. There was a 65% decrease

in the liver burden in the *pmif* RNA immunized mice at 48 h after infection (Fig. 3e). While both groups of mice developed blood-stage infection, the control mice showed a rapid increase in parasitemia and became moribund on day 19 after infection. The parasitemia in the *pmif* RNA immunized mice, by contrast, never exceeded 2% and was eliminated in all mice at day 25 (Fig. 3f). Cohorts of *pmif* or control RNA replicon immunized mice also were cured of blood-stage infection by treatment with chloroquine and re-infected 4 weeks later with *Pb*Aluc sporozoites. While both groups of mice showed a reduction parasite liver burden relative to the first infection (Fig. 3e), the *pmif* RNA immunized mice showed a 70% reduction in liver parasites when compared with the control mice and did not develop blood-stage infection (Fig. 3g, h).

These data support the conclusion that PMIF blockade by vaccination enhances the control of first infection and prevents

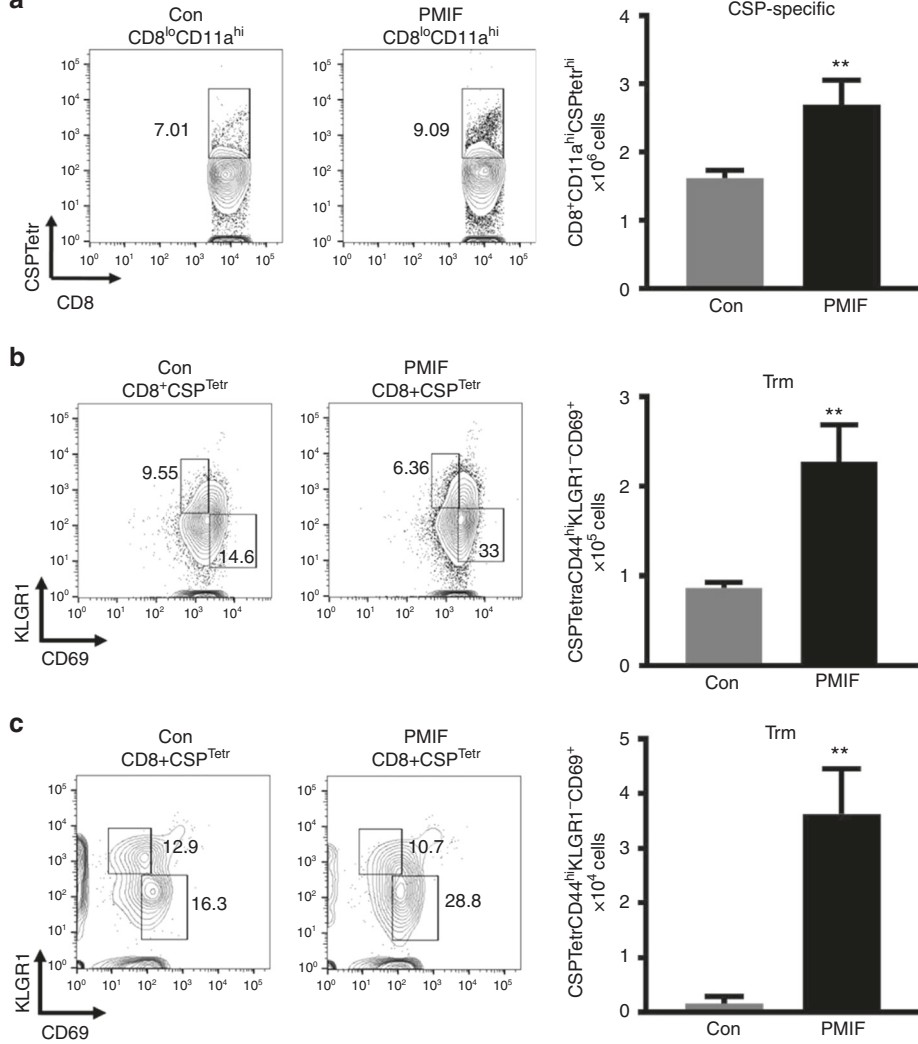

**Fig. 4** PMIF neutralization enhances the development of *Plasmodium* liver memory CD8 T cells. BALB/cJ mice immunized with replicons encoding Con RNA or PMIF RNA were challenged with 2000 *Pb*AWT sporozoites by i.v. injection. On day 7 after the first or second infection, liver immune cells were isolated and the percentage and total number of CD8 T cells assessed. **a** Representative plots and absolute numbers of CSP-specific CD8 T cells (CD8+CD11a^hi^CSPTetr^hi^) in the livers of PMIF or Con RNA immunized mice. Results are from two separate experiments. Bars represent the mean of 6 mice ± SD; **$p = 0.0022$ by Mann–Whitney test. Representative plots and absolute number of CSP-specific tissue resident memory CD8 T cells (Trm: CSPTetrCD44h^i^KLGR1−CD69+) at day 7 after the first (**b**) and second (**c**) infection. Results are from two separate experiments. Bars represent the mean of 6 mice ± SD; **$p = 0.0022$ by Mann–Whitney test

re-infection. Notably, protection was more pronounced in mice infected with *Pb*A sporozoites as these mice cleared blood-stage parasites after challenge infection and failed to develop detectable blood-stage infection after re-infection.

**A PMIF vaccine enhances liver-resident memory CD8 T cells.** As infection with *Pb*A*mif*− sporozoites is associated with an increased number of liver-resident memory CD8 T cells (Trm) (Supplementary Figure 3), we examined if immunization with *pmif* RNA also leads to enhanced Trm numbers after sporozoite infection. We hypothesized that vaccination with PMIF could have an impact in the host immunity against *Plasmodium* liver-stage by increasing the number of liver-resident memory CD8 T cells. We immunized mice with *pmif* or control RNA replicons followed by i.v. injection of 2000 *Pb*A sporozoites 1 month later and characterized the phenotype of liver CD8 T cells 7 days after infection. There was a 48% increase in the number of CSP-specific CD8 T cells in the liver (Fig. 4a) but not the spleen

(Supplementary Figure 5a) in the *pmif* RNA versus the control RNA immunized group, and examination of CD8 T cell subsets revealed a corresponding increase in CSP-specific Trm cells (Fig. 4b). Liver CD8 Trm cells directed against *Plasmodium* are long-lived[23]. To examine their development and response to a second infection, we cured immunized mice that had primary blood-stage infection by chloroquine treatment and examined CD8 Trm cell frequency 1 month later, both before and after re-infection with *Pb*AWT sporozoites. While the number of liver CD8 Trm cells decreased after 1 month in both the *pmif* and the control RNA immunized groups, there was a more than 3-fold increase in the *pmif* RNA immunized mice when compared with the control group (Supplementary Figure 5b). Seven days after re-infection with sporozoites, there was an expansion of this liver CD8 Trm population, with a 60% increase in CSP-specific CD8 Trm cells in the *pmif* RNA immunized cohort when second infection is compared to the number of Trm 1 month after the first infection (Fig. 4c). Moreover, there was an increase in the number of IFNγ-expressing CD8 Trm cells after the second

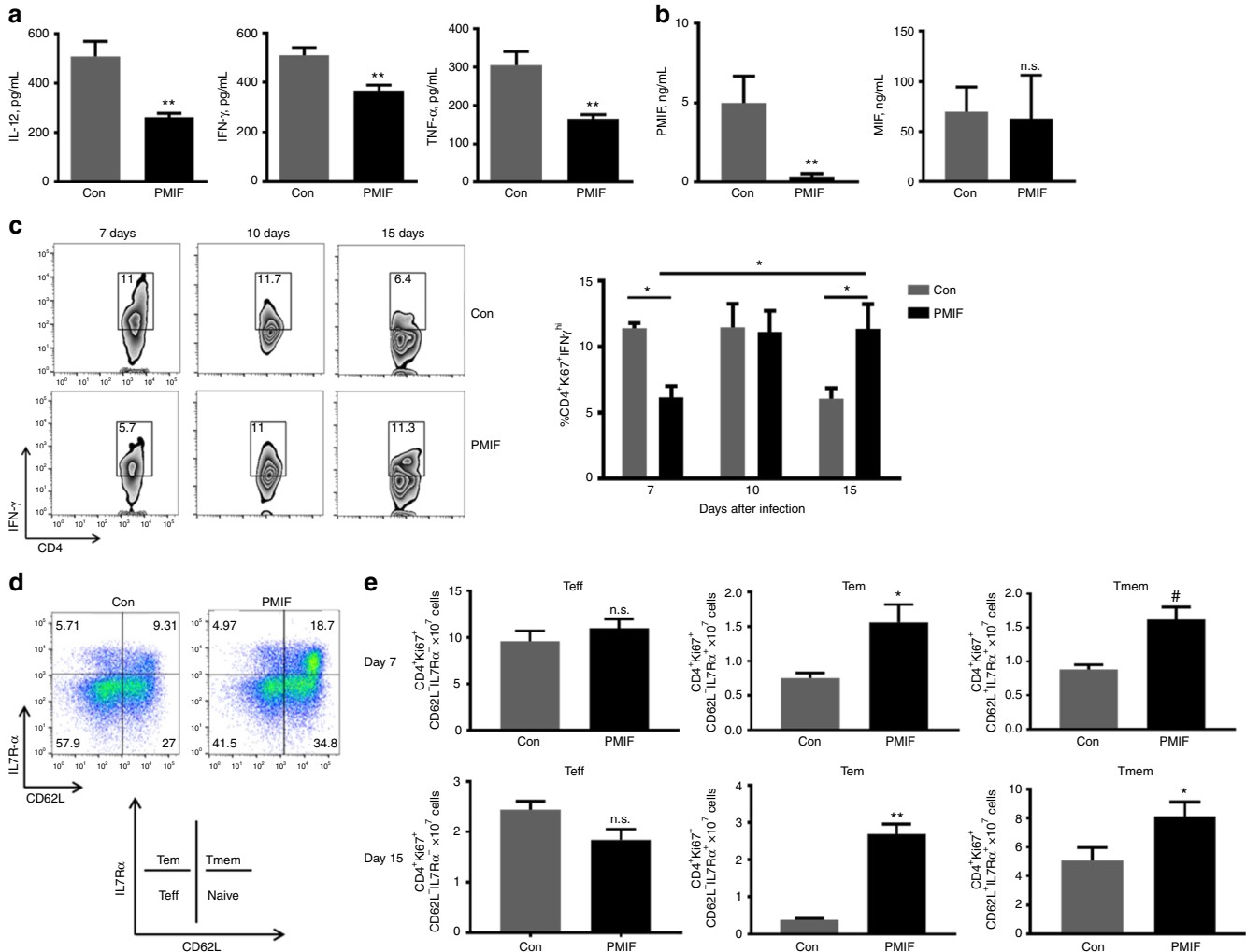

**Fig. 5** PMIF neutralization decreases inflammatory cytokine production and enhances the development of CD4 T cells into effector memory and memory precursors during blood-stage infection. **a, b** Serum levels of the indicated cytokines were detected by specific ELISA 7 days after injection of $10^6$ *Pb*AWT iRBCs in mice immunized with RNA replicons encoding Con RNA or PMIF RNA. Data are representative of two independent experiments. Bars represent the mean of 6 mice ± SD; *$p < 0.05$,**$p < 0.01$ by Mann–Whitney test. **c** On day 7, 10, and 15 after infection, splenocytes were isolated and stimulated ex vivo with iRBC lysates in the presence of Brefeldin A. Representative dot plots and frequencies of *Pb*AWT responsive CD4 T cells (Ki67+CD4+) expressing IFN-γ in spleens was detected by intracellular staining and analyzed by flow cytometry. Data are representative of two independent experiments. Bars represent the mean of 10 mice ± SD; *$p < 0.05$ by Mann–Whitney test. **d, e** Numbers of *Pb*AWT responsive CD4 T cell (Ki67+CD4+) subsets, including T effector (Teff): CD62L−IL7Rα−, T effector memory (Tem): CD62L−IL7Rα+, and T memory (Tmem): CD62L+IL7Rα+ at day 7 and 15 after infection. The contribution of each memory CD4 T cell subset is expressed relative to the total number of *Pb*AWT responsive CD4 T cells. Data are representative of two independent experiments. Bars represent the mean of 10 mice ± SD; n.s.: non-significant, *$p < 0.05$, **$p < 0.001$, #$p < 0.0001$ by Mann–Whitney test

infection in the *pmif* versus control RNA group when tested by ex vivo stimulation with sporozoites lysates (Supplementary Figure 5c). Taken together, these findings indicate that *pmif* RNA vaccination promotes a liver CD8 Trm cell response that functionally expands after re-infection.

**PMIF neutralization increases the memory CD4 T cell response.** Infection with blood-stage *Pb*A*mif−* parasites, when compared to *Pb*AWT parasites, is associated with lower circulating levels of IFN-γ and increased numbers of *Plasmodium*-responsive CD4 T cells that develop into memory precursor CD4 T cells[14]. We observed a 48% lower serum concentration of IL-12 in mice infected with *Pb*AWT iRBCs that were immunized with *pmif* versus control RNA, as well as a 30% and 45% reduction respectively, in circulating IFN-γ and TNF-α levels (Fig. 5a).

Serum concentrations of PMIF and host MIF also were measured by specific ELISA. PMIF levels were reduced by 89% in infected mice previously immunized with *pmif* RNA and, as expected, there was no alteration in the levels of circulating host MIF (Fig. 5b). While there were similar numbers of *Plasmodium*-responsive CD4 T cells in both groups of mice during acute infection (day 7), there was a 50% reduction in the percentage of CD4 T cells producing IFN-γ in the *pmif* RNA immunized group (Fig. 5c), which is consistent with reduced development of an initial inflammatory CD4 T effector population in the setting of PMIF neutralization or genetic absence. This difference in IFN-γ producing CD4 T cell population disappeared by day 10, and with resolution of infection (day 15) there was a comparative increase in the *Plasmodium*-responsive, IFN-γ expressing memory CD4 T cell population in the *pmif* RNA immunized group. These data suggest a time-dependent development and preservation of a

memory CD4 T cell response during blood-stage infection after *pmif* RNA immunization.

*Plasmodium* infection is associated with a down-regulation of the T cell survival receptor IL7Rα and an upregulation of T-bet, which are markers of the terminal differentiation of effector CD4 T cells[14,24]. Using the markers CD62L and IL7Rα to assess the phenotype of *Plasmodium*-responsive memory CD4 T cells, we observed at 7 days a 90% increase in CD4 T effector memory cells (Tem: CD62L⁻IL7Rα⁺), an 80% increase in CD4 T memory cells (Tmem: CD62L⁺IL7Rα⁺) (Fig. 5d, e), as well as a 20% reduction in the number of CD4 T cells expressing the exhaustion marker PD-1 in the *pmif* RNA versus control RNA immunized mice (Supplementary Figure 6). This observed phenotype of *Plasmodium*-responsive effector memory and memory CD4 T cells was further evidenced by measurements at 15 days of infection (Fig. 5e). Taken together, these findings support the relative preservation of a memory CD4 T cell response by PMIF neutralization during blood-stage *Plasmodium* infection.

We next examined the impact of re-infection in a cohort of *pmif* RNA immunized mice that were cured of primary blood-stage *PbA*WT infection by chloroquine treatment. Lower circulating concentrations of IFN-γ were noted after challenge infection in the *pmif* RNA immunized group when compared to controls, and this was associated with a 94% reduction in serum PMIF (Supplementary Figure 7a). There also was evidence of preservation and expansion of the CD4 T effector memory and memory cell population by 100%. (Supplementary Figure 7b). After re-infection, the number of *Plasmodium*-responsive CD4 T cells was similar in both groups but there were 40% fewer *Plasmodium*-responsive CD4 T cells producing IFN-γ in the *pmif* RNA immunized mice than in the control group (Supplementary Figure 7c). Moreover, there was a> 25% decrease in the proportion of memory CD4 T cells expressing PD-1, suggesting that the neutralization of PMIF during blood-stage infection reduces memory CD4 T cell exhaustion (Supplementary Figure 7d).

**PMIF neutralization promotes anti-PbA cellular and humoral immunity**. Immunohistochemical staining of spleens 15 days after blood-stage infection with *PbA*WT revealed an expanded and less disorganized B cell relative to T cell zone in the *pmif* RNA versus control RNA immunized mice (Supplementary Figure 8a, b). We examined the development of Tfh cells and GC B cells in spleens, first by enumerating Tfh cells (CD4⁺CD62L⁻CXR5ʰⁱPD-1ʰⁱ)[7,25] in mice infected with *PbA*WT parasites that had been immunized previously. There was a 2.5-fold increase in the number of CD4 Tfh cells when compared to infected mice immunized with a control RNA replicon. The number of pre-Tfh cells also was higher in the controls than in the *pmif* RNA immunized mice (Fig. 6a, b), supporting a maturation defect in Tfh cells in the control group. Consistent with this observation, we measured the expression of the Tfh differentiation regulator Bcl-6 and confirmed that its expression was significantly higher in the Tfh cells from the *pmif* RNA versus the control RNA immunized mice (Fig. 6a, c)[17]. Consistent with this observation, we found a significant increase in the number of GC B cells (CD19⁺CD38ˡᵒGL7⁺) and memory B cells (CD138⁻CD19⁺IgD⁻CD38ʰⁱ) during the first infection, and the difference was maintained after the second infection in *pmif* RNA immunized mice when compared with the controls (Fig. 6d, e). That *pmif* RNA immunization is associated with an improvement in the host Tfh and B cell responses was confirmed by serum antibody titers against *Plasmodium* blood- and liver-stage (CSP) antigens, and a 6-8-fold higher titer of total IgG, was observed against blood-stage and liver-stage antigen, respectively, in the

*pmif* RNA versus control RNA immunized groups (Fig. 6f, g). Taken together, these results demonstrate that immunoneutralization of PMIF reduces its detrimental effect on the development of *Plasmodium*-responsive Tfh cells, restores GC formation, and promotes a more effective cellular and humoral response against pre- and erythrocytic *Plasmodium* infection.

**PMIF vaccination elicits malaria-protective CD4 T cells**. The observation that immunoneutralization of PMIF promotes the differentiation and maintenance of a memory CD4 T cell response, improves anti-*Plasmodium* antibody responses, and prevents re-infection to blood-stage malaria prompted us to examine more closely the contribution of the adaptive and humoral responses to protective immunity. We assessed the functional significance of an augmented CD4 T cell response by adoptive transfer into naïve recipients of splenic CD4 T cells isolated from *PbA*WT-infected mice that had been immunized against *pmif* or control RNA. For this protocol, mice were sacrificed 7 days after the second infection and $2 \times 10^7$ splenic CD4 T cells (CD45.2) were CFSE-labeled and transferred into congenic CD45.1 BALB/cJ mice. The recipient mice then were infected with blood-stage *PbA*WT 3 days after adoptive cell transfer (Fig. 7a). Infection was established in recipient mice that received CD4 T cells from the control group, as evidenced by increasing parasitemia and organ parasite content, but not in mice that received CD4 T cells from the *pmif* RNA immunized donors (Fig. 7b). The phenotype of the transferred CD4 T cells also was characterized in mice euthanized at day 7 after infection. The protection conferred by the adoptive transfer of CD4 T cells from the *pmif* RNA immunized donors was associated with a higher number of proliferating CD4 T cells (CFSEˡᵒ) (Fig. 7c, d), higher levels of IFN-γ production (Fig. 7e), and reduced expression of the exhaustion marker PD-1 when compared to CD4 T cells adoptively transferred from the control group (Fig. 7f). These data indicate that the augmented CD4 T cell response that develops after *pmif* RNA immunization in infected mice is sufficient to prevent the establishment of blood-stage infection.

**PMIF vaccination promotes a protective CD8 T cell response against sporozoite infection**. Immunization with *pmif* RNA partially protects mice from sporozoite challenge and protects completely from re-infection when the initially infected mice are cured by chloroquine treatment (Fig. 3e–h). As this protection is associated with an expansion of liver CSP-specific CD8 T cells (Fig. 6), we adoptively transferred $2 \times 10^7$ liver CD8 T cells from immunized CD45.2 mice after the second infection to evaluate the functional significance of this expanded CSP-specific T cell population. Three days after adoptive transfer, we infected recipient CD45.1 Balb/cJ mice with 2000 *PbA*WT sporozoites and assessed the development of infection and the CD8 T cell response (Fig. 8a). Hepatic parasite content was significantly reduced at 48 h in mice that received liver CD8 T cells from the *pmif* RNA versus the control RNA replicon immunized hosts (Fig. 8b). Blood patency was established in recipient mice that received liver CD8 T cells from the control RNA immunized hosts but not in mice that received CD8 T cells from the *pmif* RNA immunized hosts (Fig. 8c). We euthanized the mice 7 days after infection to assess the phenotype of the transferred CD8⁺ T cells. The protection conferred by the adoptive transfer of liver CD8 T cells from the *pmif* RNA immunized hosts was associated with a higher number of proliferating CSP-specific CD8 T cells (CFSEˡᵒ) producing IFNγ (Fig. 8d, e). These data indicate that the augmented liver CD8 T cell response that develops in infected

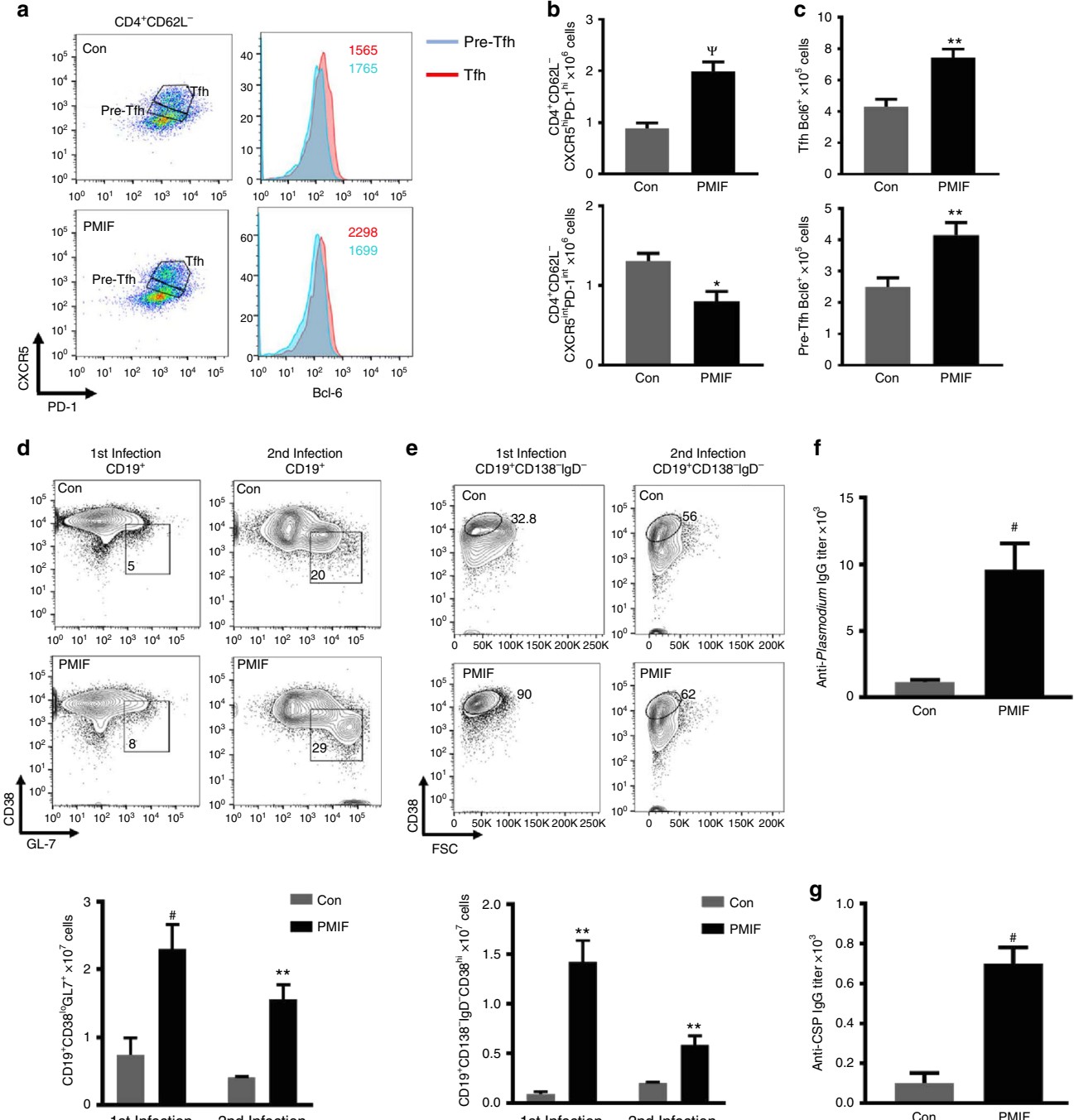

**Fig. 6** PMIF inhibition enhances the development of CD4 Tfh, plasma cells, and anti-*Plasmodium* antibody responses. BALB/cJ mice immunized with replicons encoding Con RNA or PMIF RNA were infected with 2000 *Pb*AWT sporozoites and splenocytes isolated on day 7 after infection. **a**, **b** Representative plots of absolute numbers of Tfh cells (CD4+CD62L−CXCR5hiPD-1hi) and pre-Tfh cells (CD4+CD62L−CXCR5intPD-1int). **c** Expression of transcription factor Bcl-6 in Tfh and pre-Tfh cells for both groups of mice during first *Pb*AWT infection. Results are from two separate experiments. Bars represent the mean of 6 mice ± SD; *$p < 0.05$, **$p < 0.001$, Ψ$p < 0.0001$ by Mann–Whitney test. **d**,**e** Representative plots and absolute number of germinal center (CD19+CD38loGL7+) and memory B cells (CD19+CD138−IgD−CD38hi) after the first and second infection. Results are from two separate experiments. Bars represent the mean of 6 mice ± SD; **$p < 0.001$, #$p < 0.0001$ by Mann–Whitney test. Serum titers of specific anti-*Plasmodium* blood-stage (**f**) and anti-CSP liver-stage antigen (**g**) IgG from immunized mice analyzed 1 week after the second infection with *Pb*AWT sporozoites. Data shown are from three and two independent experiments, respectively. Bars represent the mean of 10 mice ± SD; #$p < 0.0001$ by Mann–Whitney test

mice after *pmif* RNA immunization is sufficient to prevent the establishment of infection by *Plasmodium* sporozoites.

**Antibodies elicited by PMIF vaccination enhance malaria control**. We purified serum IgG from *Pb*AWT-infected mice

previously immunized with RNA encoding PMIF or GFP and tested its effect on malaria development in both the BALB/cJ and the cerebral malaria-sensitive C57BL/6J mouse strains (Supplementary Figure 9a). Administration of IgG from PMIF-immunized mice into naïve mice that were infected with *Pb*AWT provided partial protection, with a delayed rise in

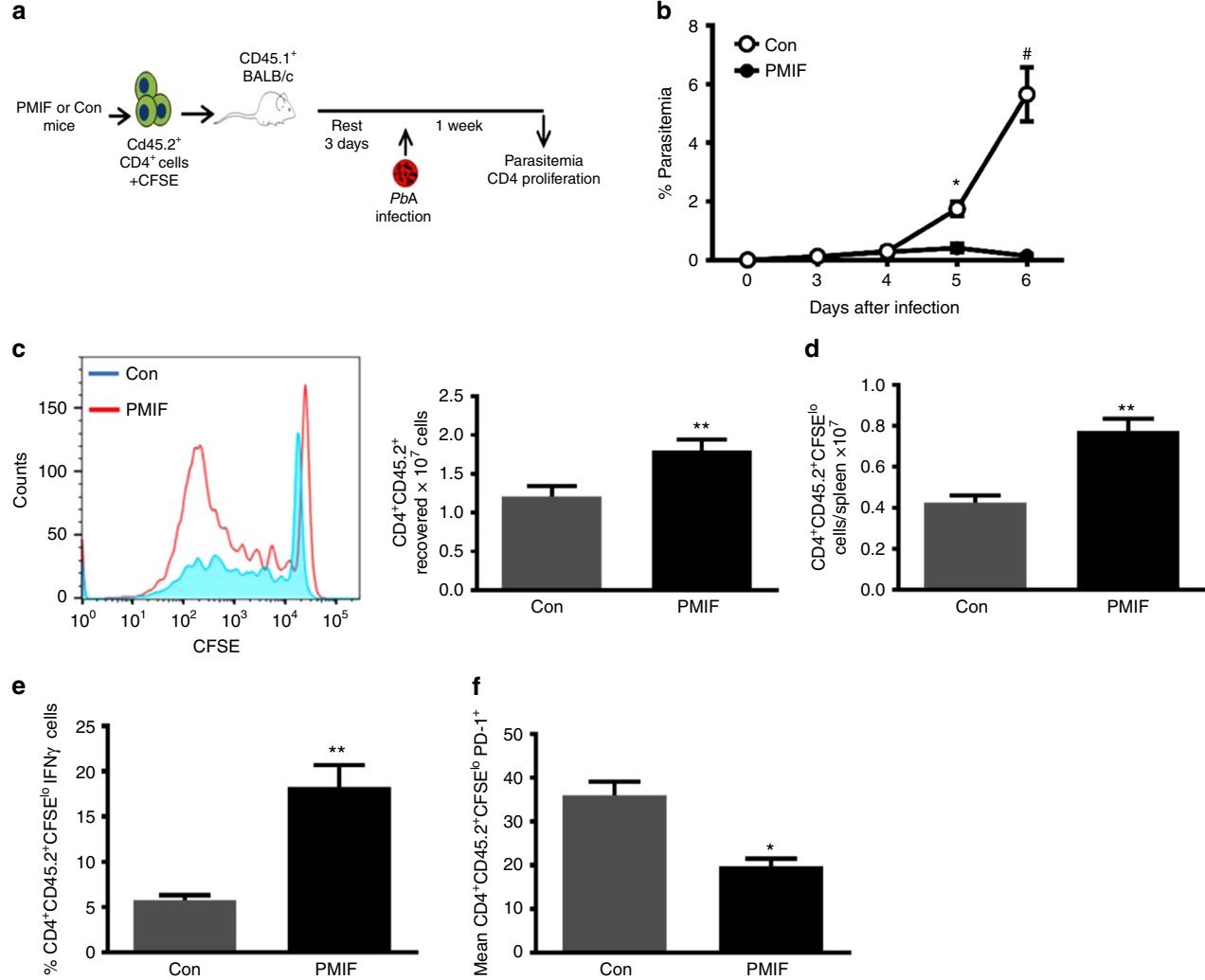

**Fig. 7** Adoptively transferred CD4 T cells from PMIF-immunized mice confer protection to challenge by iRBCs. **a** BALB/cJ mice immunized with replicons encoding Con RNA or PMIF RNA were infected with $10^6$ *Pb*AWT iRBCs and treated with chloroquine on days 7–12. Four weeks later, the mice were reinfected with $10^6$ *Pb*AWT iRBCs and splenocytes isolated 7 days after infection, incubated with chloroquine to eliminate blood-stage parasites, and labeled with CFSE. Purified CD4+CD45.2+ T cells ($2 \times 10^7$) then were transferred into naïve congenic CD45.1 BALB/cJ hosts and the mice infected 3 days later with $10^6$ *Pb*AWT iRBCs. **b** Frequency of iRBCs in mice adoptively transferred with CD4 T cells from Con (white circle) or PMIF (black circle) RNA immunized mice. Results are from two separate experiments. Bars represent the mean of 6 mice ± SD; *$p < 0.05$, #$p < 0.001$ by two-way ANOVA. **c** Representative CFSE dilution histogram of adoptively transferred CD4+ T cells (CD45.2) from donors immunized with Con or PMIF RNA and enumeration of recovered CD45.2 CD4+ T cells, and **d** proliferative response of transferred CD4 T cells into CD45.1 recipients 7 days after infection. **e** Percentage of proliferating CD45.2+ CD4+ T cells (CFSE$^{lo}$) producing IFN-γ after stimulation ex vivo with iRBC lysates in the presence of Brefeldin A. **f** Mean fluorescence intensity of PD-1 in *Pb*AWT responsive CD45.2+ CD4+ T cells (CFSE$^{lo}$) from Con or PMIF RNA immunized donors. Results are from two separate experiments. Bars represent the mean of 8 mice ± SD; *$p < 0.05$, **$p < 0.01$ by two-tailed Mann–Whitney test

parasitemia, a 30% reduction in peak parasitemia, a 30% prolongation in survival time in BALB/c mice (Supplementary Figure 9b), and a 30% reduction in lethality in C57/BL/6J mice (Supplementary Figure 9c). These data indicate that while humoral IgG was not as protective as CD4 T cell transfer, which resulted in complete protection to *Pb*AWT infection in BALB/c mice, antibody produced in the setting of PMIF vaccination was more effective in ameliorating lethality than antibody from the vaccine controls.

## Discussion
Immunity to malaria is slow to develop and the tolerance that may develop to clinical disease requires repeated infection over many years. The cellular and humoral mechanisms responsible

for the failure by the host to develop sterile immunity are not well understood but have been considered to be features of an immunosuppressive response. Several lines of evidence indicate that acute malaria can inhibit T cell development with significant impact on GC development and activity[5,26–29], and recent studies have identified that many of the same inflammatory factors that mediate severe malaria have a deleterious effect on Tfh and B cell development[7,8]. Herein we provide evidence for a mechanism by which *Plasmodium* parasites negatively regulate the adaptive response via the expression of PMIF. PMIF upregulates IL-12 and IFN-γ expression to increase the inflammatory environment of the CD4 T cell response and interfere with the differentiation of *Plasmodium*-specific CD4 T effector cells[14]. The pro-inflammatory action of PMIF during *Plasmodium* infection blocks Tfh cell development, leading to a higher frequency of Tfh

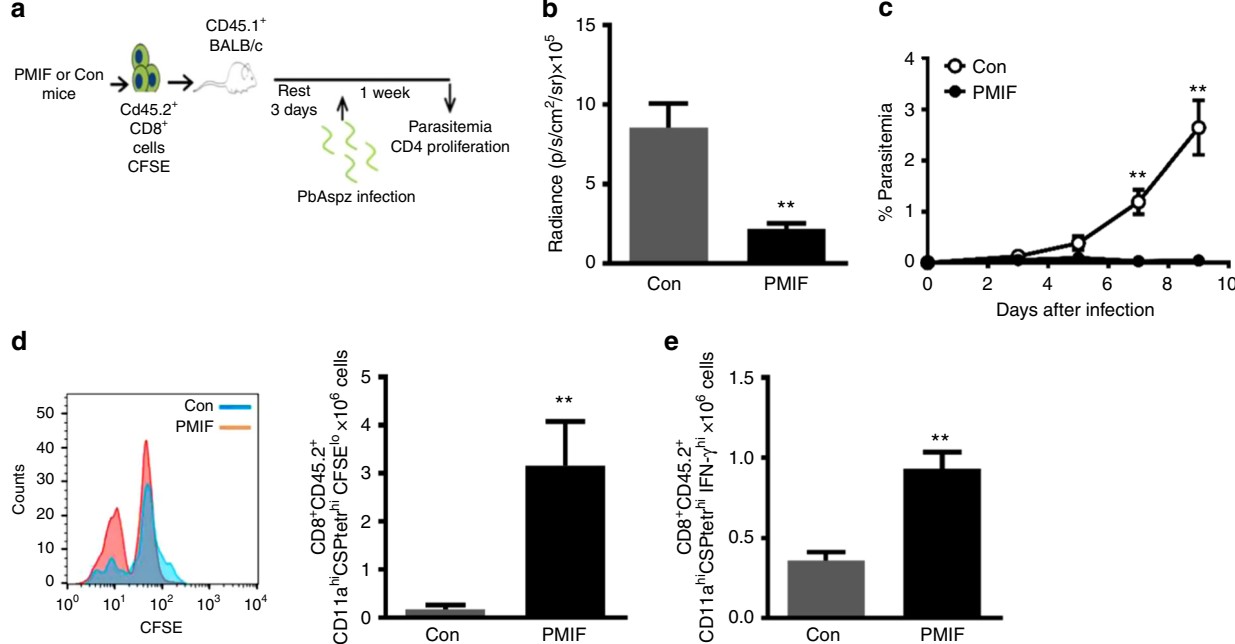

**Fig. 8** Adoptively transferred liver CD8 T cells from PMIF-immunized mice confer protection to homologous sporozoite challenge. **a** BALB/cJ mice immunized with replicons encoding Con RNA or PMIF RNA were infected with 2000 *Pb*AWT sporozoites and cured by 6 days of chloroquine treatment (days 7–12). Four weeks later, the mice were reinfected with 2000 *Pb*AWT sporozoites and T cells from liver isolated 7 days after infection, incubated with chloroquine to eliminate residual blood-stage parasites, and labeled with CFSE. Purified CD45.2$^+$ CD8 T cells ($2 \times 10^7$) then were transferred into naïve congenic CD45.1 BALB/cJ hosts and the mice infected 3 days later with 2000 *Pb*AWT sporozoites. **b** Luminescence values of infected mice and **c** parasitemia in mice adoptively transferred with liver CD8 T cells from Con RNA (white circle) or PMIF RNA (black circle) immunized mice. Results are from two separate experiments. Bars represent the mean of 6 mice ± SD **$p < 0.01$ by two-way ANOVA. **d** Representative CFSE dilution histogram of adoptively transferred (CD45.2) CD8 T cells from Con RNA or PMIF RNA immunized donors and enumeration of recovered CD45.2 CD8 T cells. **e** Number of proliferating CD45.2 CD8 T cells (CFSE$^{lo}$) producing IFN-γ after stimulation ex vivo with CSP peptide in the presence of Brefeldin A. Results are from two separate experiments. Bars represent the mean of 6 mice ± SD;**$p < 0.01$ by two-tailed Mann–Whitney test, error bars denote ±SD

precursors expressing Th1-type molecules such as T-bet and IFNγ, and a lower frequency of pre-Tfh cells expressing Bcl-6. This interference in Tfh cell maturation by PMIF is associated with a detrimental effect on the induction of GC and B cell responses (Supplementary Figure 11).

These findings prompted us to examine if immunization against PMIF could reduce its deleterious action during preerythrocytic or erythrocytic stage infection. While initial sporozoite infection produced patent infection in immunized hosts, parasitemia was significantly attenuated. Moreover, mice that were cured of infection but received a second sporozoite challenge were fully protected against re-infection. This protection was associated with enhanced development of liver CSP-specific CD8 T cells that were predominantly of the Trm phenotype. The protective action of liver-resident CD8 T cells was confirmed by adoptive transfer, which fully protected naïve hosts from developing detectable parasitemia after sporozoite infection. When infection was initiated with *Plasmodium* blood-stage malaria, prior immunization against PMIF reduced the expression of IL-12 and IFN-γ, resulting in an increased number of *Plasmodium*-responsive CD4 T cell memory precursor cells, and an expansion of CD4 T cell effector memory and memory population. A disorganization of GC architecture has long been noted to occur in human and experimental malaria, and is associated with an impairment in an effective antibody response[7,30,31]. PMIF immunization led to a preservation of splenic GC architecture and B cell zonal expansion, an increase in the number of CD4 Tfh cells and GC B cells, and a higher anti-*Plasmodium* antibody titer. Adoptive transfer of the CD4 T cells that develop in PMIF-immunized mice during *Plasmodium* blood-stage infection also conferred full protection to blood-stage infection in naïve hosts.

Upon transfer, these CD4 T cells showed enhanced *Plasmodium*-specific proliferation and IFN-γ production, and reduced exhaustion.

Taken together, these findings demonstrate remarkable protection in a lethal murine model of malaria initiated by sporozoite or blood-stage infection. These data also affirm the role of the *Plasmodium*-encoded factor PMIF in actively interfering with the adaptive immune response by a pro-inflammatory mechanism involving engagement of the host MIF receptor[14] to suppress the differentiation of memory CD4 and CD8 T cell subsets. We suggest that the marked protection observed by PMIF immunization may prompt consideration of this antigen as a vaccine candidate, either as a standalone immunogen or in combination with other *Plasmodium* antigens, where it could act to ensure the development and maintenance of adequate memory responses in endemic settings. It is notable that closely homologous MIF orthologs have been described in other parasitic protozoan and helminthic species[32–35]. Conceivably, this family of evolutionary conserved proteins provides a generalized mechanism by which parasites interfere with the adaptive response to maintain persistence in the mammalian host and completion of their life cycles.

## Methods

**Mice, parasites, and cell lines**. All animals were maintained in specific pathogen-free facility at Yale Animal Resource center (YARC). All animal procedures followed federal guidelines and were approved by the Yale University Animal Care and Use Committee (UACUC), approval number 2017-10929. Females. BALB/cJ (CD45.2, CD45.1) and C57/BL6J mice between 6 and 10 weeks of age were used and purchased from Jackson Laboratories. Cryopreserved wild-type *P. berghei* ANKA (MR4) (*Pb*AWT) and *Pb*Amif− parasites[13] were passaged once through Swiss Webster mice before infection in experimental animals. Eight to ten-week-

old mice were infected intraperitoneally (i.p.) with $10^6$ PbA iRBCs and the ensuing parasitemia assessed by enumeration of Giemsa-stained thin blood smears, flow cytometry using a Tri-color method (TCM)[36] and by qPCR for splenic PbA 18S rRNA[37]. For sporozoite infection, PbAWT, PbAmif− (Leiden Malaria Group) or PbAWT-GFP (MR4) parasites were cycled between Swiss Webster mice and Anopheles stephensi mosquitoes. Salivary gland sporozoites were extracted from infected mosquitoes on day 19 post-blood meal infection. BALB/cJ mice were infected by tail i.v. injection of 2000 PbA sporozoites and blood patency was monitored beginning day 3 by blood smear, flow cytometry and when infected with PbAWT-GFP parasites, the liver burden was monitored using an IVIS imaging system.

**Adoptive transfer of splenic CD4 T cells and liver CD8 T cells**. Splenocytes or liver lymphocytes were isolated from vaccinated and infected CD45.2$^+$ BALB/cJ mice at day 7 after the second infection and incubated with 10 μM chloroquine for 2 h at 37 °C. Splenic CD4 T and liver CD8 T cells were purified with anti-CD4 or anti-CD8 microbeads (CD4 (L3T4) and CD8a (Ly-2) Myltenyi Biotech) according to manufacturer's protocol. Purified CD4 T or CD8 T cells were labeled with 5 μM CFSE (Life Technologies) and $2 \times 10^7$ cells transferred i.v. into recipient CD45.1$^+$ BALB/cJ mice. Recipient mice were infected with $1 \times 10^6$ iRBCs or 2000 sporozoites 3 days after transfer and parasitemia monitored daily. At day 7 post-infection for iRBC infection or at day 9 for sporozoites infection, mice were sacrificed and donor CD4 or CD8 CD45.2$^+$ T cells recovered, quantified, and proliferation assessed by CFSE dilution.

**Flow cytometry**. For splenic cell profiling assessment, spleens were harvested at the indicated days after infection, homogenized, and passed through a 70 μm strainer to obtain single-cell suspensions. Red blood cells were lysed with ACK lysis buffer and splenocytes were stained using fluorophore-labeled antibodies (BD Biosciences) directed against Ki67 as a surrogate marker of malaria-specific cells[36], CD3, CD4, IL-7Rα, CD62L, PD-1, CD45.2, and IFN-γ for memory CD4 T cells, CXCR5, PD-1, CD62L, T-bet, Bcl6, and IFN-γ for Tfh or pre-Tfh cells, and B220, CD19, CD138, IgM, Gl7, CD38, and IgD for the B cell lineage. For intracellular cytokine staining, cells were stimulated ex vivo by co-culture of naïve CD45.1 splenocytes with iRBC lysates or recombinant PMIF[14] for 5 h in the presence of 1 μg/mL Brefeldin A (BD Bioscience). To assess the development of memory immune responses in mice infected with sporozoites, perfused livers were harvested at the indicated days after infection, homogenized, and the lymphocytes isolated by centrifugation at 500g for 15 min at RT using a 35% Percoll gradient; splenocytes were isolated as described previously. Liver and spleen lymphocytes were stained first with CSP-specific tetramers (NIH Tetramer Facility) for 20 min at 4 °C, followed by antibodies directed against CD3, CD8, CD69, KLGR1, CD44, CD62L, CD45.1, and IFNγ (Biolegend) refer to Supplementary Table 1 for antibodies source and concentration. For intracellular cytokine staining, cells were stimulated ex vivo by co-culture of naïve CD45.1 splenocytes with sporozoite lysates for 5 h in the presence of 1 μg/mL of Brefeldin A before labeling with the specified antibodies from Biolegend. For each staining, the corresponding Fluorescence Minus One (FMO) controls[38] were performed (see Supplementary Figure 10 for representative analysis). Stained cells were analyzed on an LSR II flow cytometer (BD Bioscience) and data processed with FlowJo software (TreeStar).

**RNA synthesis, nanoparticle formulation, and vaccination**. The synthesis of the self-amplifying RNA (replicon) from a modified alphavirus encoding pmif or control RNA was performed as previously described[39]. The control RNA comprised green fluorescent protein (gfp)[39], or in experiments requiring luminescent Plasmodium detection, secreted placental alkaline phosphatase (seap)[40]. No differences in background host responses were noted between the two different controls (71,74). Briefly, the codon optimized sequences were inserted into the subgenomic reading frame of a modified DNA plasmid encoding the self-amplifying RNA[41]. The plasmid was linearized by restriction enzyme digestion immediately following the 3′-end of the self-amplifying RNA sequence. Linearized DNA was transcribed into RNA with the MEGAscript T7 Transcription Kit (Life Technologies). Transcripts were purified by precipitation in the presence of 2.8 M lithium chloride, capped using the ScriptCap m7G Capping System (CellScript), and re-precipitated with lithium chloride. A cationic nanoemulsion was prepared and characterized for particle size, RNase protection, and endotoxin as previously described and allowed to complex for at least 30 min before immunization[42]. Female BALB/cJ mice (8–10 weeks old) were injected i.m. in hind limbs on day 0 and on day 21 with 15 μg of the pmif or control RNA expressing replicon. Blood was collected in both groups of animals by orbital bleeding of anesthetized mice at day 0 pre-immunization, and at day 14 and day 35 post-immunization to titer the anti-PMIF specific antibody response.

**Western blotting and anti-PMIF and anti-Plasmodium ELISA**. For the detection of anti-PMIF antibody, microtiter plates (Nunc) were coated with 100 ng/mL of recombinant PMIF or mouse MIF, incubated overnight, washed, and blocked with assay diluent (eBioscience) for 1 h. Mouse sera were serially diluted and added to wells for 2 h. Antibody binding to PMIF or mouse MIF[14] was measured by addition of HRP-labeled goat anti-mouse antibodies (1/1000, Southern Biotech). To

measure the titer of anti-Plasmodium-specific IgG, mouse sera were collected on day 7 after infection and the microtiter plates were coated with 1 μg/mL of PbAWT iRBC lysates as antigen[43]. Reciprocal endpoint titers were calculated as the reciprocal of the dilution at which the O.D. was twice background observed in uninfected mice.

For western blotting, 100 ng of recombinant PMIF or mouse MIF[29] were electrophoresed by SDS-PAGE in Tris-glycine gel (Bio-Rad) and then transferred to a PVDF membrane (Millipore). The PMIF and mouse MIF proteins were probed for serum antibody responses by the addition of PMIF or GFP (control) immune serum (1/1000) and detection with rabbit anti-PMIF antibodies or goat anti-mouse MIF antibodies (1/1000, Santa Cruz Biotech.) as previously described[14]. The signal was developed by ECL HRP substrate and the membrane exposed to a film (Amersham).

For the study of PMIF expression by merozoites, iRBC were lysed with saponin lysis buffer (0.03% saponin in PBS) and merozoites pelleted by centrifugation at 1500×g for 5 min at 4 °C. Merozoites were lysed with lysis buffer (25 mM Tris–HCl pH 7.6, 150 mM NaCl, 1% NP-40, 1% sodium deoxycholate, 0.1% SDS) for 30 min in ice[44]. The lysates were centrifugated at 15,000×g for 15 min (4 °C) and the supernatant (cytosolic fraction) and pellet (membrane fraction) were separated and mixed with SDS-PAGE loading buffer (BioRad). Protein extracts were separated by gel electrophoresis and transferred to PVDF membrane (Millipore). The membrane was probed with rabbit anti-PMIF[14] (1/1000) or anti-MSP-1 IgG (1/1000) MRA-667, MR4 ATTC, and after incubation with anti-rabbit IgG-RD800 (1/15000, Li-Cor), the membrane was imaged with an Odyssey system (Li-Cor). All uncropped Western Blot scans are included in Supplementary Figure 12.

To measure the titer of anti-Plasmodium-specific IgG[14], sera were collected on day 7 after infection and microtiter plates coated with 1 μg/mL of PbA iRBC lysates or 100 μg/mL of CSP peptide as antigen[43]. Reciprocal endpoint titers were calculated as the reciprocal of the dilution at which the O.D. was twice background observed in uninfected mice.

**Histology**. To examine GC architecture, spleens were removed from mice and fixed in 10% formalin (Sigma-Aldrich). For immunostaining, paraffin sections where deparaffined, hydrated and treated with antigen retrieval buffer (Dako). The sections then were permeabilized by 4 min treatment with Triton X-100 (0.1% in PBS) and incubated for 90 min with blocking buffer (Dako) followed by an overnight incubation with primary antibodies: rat anti-mouse B220 and hamster anti-mouse CD3e (1/1000, BD Pharmingen). For immunohistochemistry, antibodies were detected with AP-conjugated goat anti-Armenian hamster IgG or HRP-conjugated donkey anti-rat IgG (1/10000, Jackson ImmunoResearch Laboratories). HRP was reacted with DAB (Peroxidase Substrate Kit; Vector), and alkaline phosphatase with Fast Blue/Napthol AS-MX (Sigma-Aldrich). Levamisole (Sigma) was used to block endogenous alkaline phosphatase activity and slides were mounted in Crystal Mount (Electron Microscopy Sciences). Sections were viewed under a Nikon Microphot FXA light microscope and photographs were taken with a Spot Insight Camera, using 10× and 40× objectives, then analyzed using Spot Advanced software (Diagnostic Instruments) and ImageJ (NIH).

**Statistical analysis**. All data were expressed as a mean ± SD of at least two independent experiments. Differences in parasitemia involving repeated measurements were analyzed using a two-way ANOVA. Mouse survival times were analyzed by the Mantel-Cox log-rank test. All other data were first tested for Gaussian distribution of values using a D'Agostino-Pearson normality test. The statistical significance of differences was assessed using the Mann–Whitney U test for non-parametric data distribution or Student's t-test for parametric data. All statistical analysis was performed using Software Prism v.6.0, (GraphPad). p Values of less than 0.05, 0.01, or 0.001 were used to indicate statistical significance.

**Ethics approval**. All animal procedures followed federal guidelines and were approved by the Yale University Animal Care and Use Committee, approval number 2017-10929

**Data availability**. The data that support the findings of this study are available from the corresponding author upon reasonable request.

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

## Acknowledgments

This work was funded by National Institutes of Health Grants AI 5R01-51306-05, AI 2R01-042310-12, and Novartis Vaccines, Inc. This study was supported by the Deutsche Forschungsgemeinschaft grant SFB1123/A03 to J.B. We thank Michelle Chan and Nisha Chandler for coordinating the delivery of formulations for the animal studies. We thank MR4 for providing us with malaria parasites provided by Mark F. Wisser, Andy Waters, and Victor Nussenzweig.

## Author contributions

A.B.G., A.G., J.B. and R.B. conceived and designed the experiments. A.B.G., E.S., V.E. and T.S. performed the experiments. L.B., A.H., G.O. and J.U. contributed with RNA vector and vaccine production. C.J.J., E.F. and K.A. provided reagents. A.B.G. and R.B. analyzed the data and wrote the paper.>

## Additional information

**Competing interests:** Yale University and Novartis AG have filed a joint patent application describing the potential utility of a *pmif* encoding RNA replicon. R.B. and A.G. are co-inventors on this application. The remaining authors declare no competing interests.

