## [Peer Review File · Nature Communications]

Reviewers' comments:

Reviewer #2 (Remarks to the Author):

In this study Bucala and colleagues demonstrate the importance of the Plasmodium ortholog of Macrophage Migration Inhibitory Factor (PMIF) in modulating host CD4 T cell and B cell response, allowing parasite escape from host immune responses. When infecting mice with a genetically manipulated PMIF^{-/-} *P. berghei*, parasitemia and lethality are delayed. Importantly, vaccination with PMIF RNA amplicons induced high levels of anti-PMIF abs and result in strong protection of mice infected with WT parasites. The main hypothesis is that PMIF prevents development of Thf cells by promoting a strong response of Th1 lymphocytes accompanied by high levels of IL-12, IFN γ , and TNF α . As a consequence, in immunized mice development of Thf and B memory cells is augmented and formation of germinal centers (GCs) is restored and levels of parasite-specific antibodies is augmented. As a consequence one would imagine that antibody response to the parasite is augmented and control of parasitemia becomes efficient. In addition, immunized mice become more resistant to sporozoite challenge, due to enhanced migration of CD8⁺ T cells to the liver.

There are some points that need to be clarified:

- 1) Levels of antibodies comparing mice infected with WT vs PMIF⁻ parasites should be moved to main figure.
- 2) Since the hypothesis of host protection in PMIF^{-/-} mice is based on recovery of GCs, one would assume that it is the increase of parasite specific antibodies that mediates protection in immunized mice. The alternative hypothesis is that abs from immunized mice could be directly targeting the parasite blood stages. Is PMIF expressed on surface of *P. berghei* blood stages or infected erythrocytes?
- 3) It is curious that in CD4⁺T cell transfer from immunized mice, such a fast response and control of parasitemia. This is unexpected that most likely there was no time to produce high levels of antibodies specific to the parasite or to PMIF. This suggests that protection is mediated directly by CD4⁺ T cells and not by antibodies. The levels of parasite-specific and PMIF-specific abs in the cell transfer experiments need to be evaluated. Does passive transfer of anti-PMIF abs protect mice from infection with *P. berghei* blood stage?
- 4) One would assume that PMIF acts on host MIF receptor. Anti-PMIF do not cross-react with MIF and MIF levels are still high in infected mice. Why MIF is not sufficient to stimulate Th1 lymphocytes, prevent differentiation of Thf, formation of GCs and anti-parasite abs production? Does in vivo neutralization of host MIF results in a better outcome of *P. berghei* infection? In other words, is there an additive effect of PMIF and MIF?
- 5) Another curious finding is that vaccination of with PBMIF also results in enhanced CD8 response and protection to sporozoite challenge. There are so few sporozoite that reach the hepatocytes that it is hard to believe that they will produce enough PMIF to modulate host immune response. Do sporozoites express PMIF. What is the mechanism by which PMIF enhances the CD8 response to sporozoites?

Reviewer #3 (Remarks to the Author):

The manuscript titled 'Neutralization of the Plasmodium-encoded MIF Ortholog Confers Protective Immunity against Malaria Infection' by Bucala and colleagues examines how the analogue of a mammalian protein, macrophage migration inhibitory factor (PMIF) expressed by Plasmodium

berghei might suppress anti-malarial immunity in mice. The authors neutralize PMIF in *P. berghei* ANKA (PbAwt) infected mice to show that adaptive immune responses were improved, to help control the current and future infections better. However, in my opinion, some key conceptual and interpretational flaws in this study compromises its validity. Since it is not feasible to go through each and every one of such instances, please find below a few examples that illustrate my concern:

1. One of the main findings of the paper is that PMIF down-modulates the immune responses against Plasmodium, thus abetting the infection. Yet the authors admit that PMIF deficient PbA (PbAmif-) parasites are controlled to the same extent as PbAwt parasites (Page5, line 7-8), unlike their own findings in the case of MIF knocked out Leishmania (ref 1). The cited reference (#14) does not seem to show the indicated data on perusal, either. This seemed to contradict the proposed contribution of PMIF in modulating protective immunity against blood-stage malaria.
2. The authors test for the role of PMIF in PbAwt infection, by trying to neutralize PMIF through immunization of the host with a self-amplifying PMIF gene encoded RNA. The idea is that anti-PMIF antibodies generated here would neutralize PMIF made by the blood-stage plasmodia. This is probably because PMIF is secreted/ surface expressed by blood stage plasmodia (although the protein does not have a known signal peptide/ transmembrane domain as per the protein/gene annotation, PBANKA_1444000 on www.PlasmoDB.org). The authors immunized the mice and show that pMIF specific IgG and CD4 T cells responses were elicited (Fig S3a-c). And these mice when challenged with PbAwt, but not Pbmif- parasites showed better control of the infection (FigS3h). They interpret this as a result of the anti-PMIF IgG generated by immunization, neutralizing the PMIF made by PbAwt. However, generating antibodies against a blood-stage parasite protein (PMIF here), leading to the control of the infection is merely the rediscovery of the concept of immunization. The anti-PMIF antibodies may be killing the parasite by targeting its PMIF, similar to what immunizing against any other surface expressed protein like TRAP or PbT may achieve. The authors do not demonstrate actual neutralization of PMIF, nor differentiates the process from a direct killing of the parasite. This alternate interpretation gains more impetus in the context of a similar control of PbAwt and Pbmif- parasites in a primary infection (as mentioned in (1), above) and the significant boost in anti-PMIF IgG observed in PbAwt challenge of PMIF immunized mice (Figure S3B). Given this data set is central to the overall findings of the paper, I have my doubts on the general interpretations of the study.

Malaria itself is known to cause severe immunosuppression in mice. Hence, immunizing against and controlling the infection better is expected to improve surrogate determinants of stronger immune responses. The authors have failed to distinguish this possibility from the proposed neutralization of PMIF to directly improve host immunity.

Though I feel that its flaws do not completely discredit the suggestion of the manuscript that PMIF may be a direct immunomodulator as indicated in figure 1, the somewhat non-critical evaluations of the results that followed might disqualify the mechanism attributed to it.

1 Holowka, T. et al. Leishmania-encoded orthologs of macrophage migration inhibitory factor regulate host immunity to promote parasite persistence. *FASEB J* 30, 2249-2265, doi:10.1096/fj.201500189R (2016).

Authors' Response to Critique of:

NCOMMS-17-23430, "Neutralization of the *Plasmodium*-encoded MIF Ortholog Confers Protective Immunity against Malaria Infection".

The Authors' response is in *italicized blue font* below each Reviewer's point. Revisions that appear to the main text are indicated by **blue font**. (Note that Reviewer designations are as provided to authors and the absence of a "Reviewer #1").

Reviewer #2

In this study Bucala and colleagues demonstrate the importance of the Plasmodium ortholog of Macrophage Migration Inhibitory Factor (PMIF) in modulating host CD4 T cell and B cell response, allowing parasite escape from host immune responses. When infecting mice with a genetically manipulated PMIF^{-/-} *P. berghei*, parasitemia and lethality are delayed. Importantly, vaccination with PMIF RNA amplicons induced high levels of anti-PMIF abs and result in strong protection of mice infected with WT parasites. The main hypothesis is that PMIF prevents development of Thf cells by promoting a strong response of Th1 lymphocytes accompanied by high levels of IL-12, IFN γ , and TNF α . As a consequence, in immunized mice development of Thf and B memory cells is augmented and formation of germinal centers (GCs) is restored and levels of parasite-specific antibodies is augmented. As a consequence, one would imagine that antibody response to the parasite is augmented and control of parasitemia becomes efficient. In addition, immunized mice become more resistant to sporozoite challenge, due to enhanced migration of CD8⁺ T cells to the liver.

There are some points that need to be clarified:

1) Levels of antibodies comparing mice infected with WT vs PMIF^{-/-} parasites should be moved to main figure.

In revision, we have moved the original Supplementary Figure 1 showing these data to the main Figure 1 (new Fig. 1c).

2) Since the hypothesis of host protection in PMIF^{-/-} mice is based on recovery of GCs, one would assume that it is the increase of parasite specific antibodies that mediates protection in immunized mice. The alternative hypothesis is that abs from immunized mice could be directly targeting the parasite blood stages. Is PMIF expressed on surface of *P. berghei* blood stages or infected erythrocytes?

*All members of the MIF cytokine superfamily lack a signal peptide or transmembrane domain and to the extent that individual members have been studied, are released from cells by non-classical mechanisms (J Immunol. 182, 6896-6906). Previously published data from the group of Andy Waters, who first published on PMIF function using recombinant protein and PMIF gene knockout *P. berghei*, indicate that PMIF is secreted into the infected erythrocyte and released upon schizont rupture (manuscript ref. #13, see Fig. 4). Anti-PMIF antibody labeling or tagged protein expression also does not reveal cell surface PMIF (manuscript ref. #13). We confirmed schizont release in our own studies by PMIF specific ELISA and previously reported that circulating PMIF levels correlate with parasitemia and with inflammatory cytokine levels in infected mice and malaria patients (manuscript ref. #14, Fig. 2 therein, and data not shown).*

To additionally address this concern, we have performed experiments to verify surface non-expression of PMIF. We incubated uninfected erythrocytes (niRBC) or PbAWT infected erythrocytes (PbAwT iRBC) with Control IgG (Ctrl IgG), anti-PbMIF IgG, or anti-PbMSP-1. We lysed the erythrocytes, separated the proteins by SDS gel electrophoresis, and performed Western blotting to detect the retention of IgG (50 KDa heavy and 25 KDa light chains) as a means to demonstrate surface PMIF. As shown in the left panel below (a), no IgG was detectable, which is consistent with previously published results that PMIF and MSP-1 are not expressed on infected erythrocyte cell surfaces. Moreover, the

right panel shows the expression of both PMIF (12.5 KDa) and PMSP (19 KDa) in parasite lysates after rupture of PbAWT infected erythrocytes by osmotic shock.

In revision, we also provide a scheme illustrating the proposed mode of action of immunization against PMIF (**new Suppl Fig. 11**). Anti-PMIF antibodies produced in vaccinated mice block the activity of released PMIF, which reduces myeloid cell activation by limiting the binding between PMIF and the host MIF receptor CD74. Reduced PMIF activity decreases host production of pro-inflammatory cytokines, diminishes Th1 responses, and promotes Tfh responses and the recovery of GC.

3) It is curious that in CD4+T cell transfer from immunized mice, such a fast response and control of parasitemia. This is unexpected that most likely there was no time to produce high levels of antibodies specific to the parasite or to PMIF. This suggests that protection is mediated directly by CD4+ T cells and not by antibodies. The levels of parasite-specific and PMIF-specific abs in the cell transfer experiments need to be evaluated. Does passive transfer of anti-PMIF abs protect mice from infection with *P. berghei* blood stage?

We agree with this interpretation by the Reviewer. We point out that the transferred CD4 T cells were purified from secondary (re-) infected mice and thus are enriched for *Plasmodium*-specific cells. In revision, we have formally tested for the protective action of anti-PMIF antibody alone by transferring immune serum from vaccinated and secondary infected mice into naïve mice of two strains: the BALB/c model under study and the cerebral-malaria sensitive C57BL/6J mice (**new Suppl. Fig. 9**). In both mouse models of infection, only partial protection was observed from ensuing parasitemia and lethality, supporting the important role for CD4+ T cell immunity.

4) One would assume that PMIF acts on host MIF receptor. Anti-PMIF do not cross-react with MIF and MIF levels are still high in infected mice. Why MIF is not sufficient to stimulate Th1 lymphocytes, prevent differentiation of Thf, formation of GCs and anti-parasite abs production? Does in vivo neutralization of host MIF results in a better outcome of *P. berghei* infection? In other words, is there an additive effect of PMIF and MIF?

Published Biacore data indicate the PMIF binds to the host MIF receptor CD74 with a comparable K_d as mammalian MIF, and similarly induces pro-inflammatory activation in target macrophages (manuscript ref. #14). There are some published data that mammalian MIF has Th1 actions in model systems; however, we suspect that it is the contextual features of PMIF production and action (e.g., at sites of erythrophagocytosis, antigen presentation, and T cell collaboration) that is critical to explain

the present set of observations. MIF and PMIF levels are both elevated in infection (manuscript ref. #14); however, blood compartment levels may not reflect PMIF action and MIF receptor action at sites of T cell differentiation.

To the author's point, Mif-KO mice show modest protection from experimental Plasmodium infection, with evidence of enhanced numbers of activated CD4+ T cells (manuscript ref #22; and J. Immunol 186, 6271-6279).

5) Another curious finding is that vaccination of with PBMIF also results in enhanced CD8 response and protection to sporozoite challenge. There are so few sporozoite that reach the hepatocytes that it is hard to believe that they will produce enough PMIF to modulate host immune response. Do sporozoites express PMIF. What is the mechanism by which PMIF enhances the CD8 response to sporozoites?

We show sporozoite protection data because this is an "obvious" early question to pose for any vaccination approach. In our original text, we suggested the possibility of a potentially distinct mechanism by which PMIF immunization protects from sporozoite challenge (i.e., by interfering with PMIF-dependent liver stage development) by citing recent work by Miller et al. (manuscript ref. #15). In that study, the authors provide evidence for a novel role for sporozoite-expressed PMIF in liver stage development,

*In revision, we provide a **new Suppl. Fig. 3** that shows that PbAmif- parasites have impaired liver-stage development (in agreement with the initial observations of Miller et al., manuscript ref. #15), and that this impairment is related to an increase on the number of sporozoite-specific liver resident CD8+ T cells. Mammalian MIF is known to protect cells from p53-dependent apoptosis (PNAS 99:345-350; PNAS 100:9354-9359) and our ongoing studies suggest that PMIF also exerts this action to prolong the survival of infected cells. We hypothesize that hepatocytes infected with PbAmif- parasites become apoptotic due to unopposed p53 activation; the Plasmodium proteins then become exposed to resident hepatic CD8+ T cells. By analogy, PMIF-vaccinated mice would show impaired development of PbAWT parasites. As we are continuing to develop this line of investigation experimentally, we prefer to reserve these data for a subsequent publication, however we can provide these additional data if deemed necessary by the Reviewer.*

Reviewer #3

The manuscript titled 'Neutralization of the Plasmodium-encoded MIF Ortholog Confers Protective Immunity against Malaria Infection' by Bucala and colleagues examines how the analogue of a mammalian protein, macrophage migration inhibitory factor (PMIF) expressed by Plasmodium berghei might suppress anti-malarial immunity in mice. The authors neutralize PMIF in P. berghei ANKA (PbAwt) infected mice to show that adaptive immune responses were improved, to help control the current and future infections better. However, in my opinion, some key conceptual and interpretational flaws in this study compromises its validity. Since it is not feasible to go through each and every one of such instances, please find below a few examples that illustrate my concern: 1. One of the main findings of the paper is that PMIF down-modulates the immune responses against Plasmodium, thus abetting the infection. Yet the authors admit that PMIF deficient PbA (PbAmif-) parasites are controlled to the same extent as PbAwt parasites (Page5, line 7-8), unlike their own findings in the case of MIF knocked out Leishmania (ref 1). The cited reference (#14) does not seem to show the indicated data on perusal, either. This seemed to contradict the proposed contribution of PMIF in modulating protective immunity against blood-stage malaria.

We submit that the tempo and pathology of Leishmania major infection differ from that of Plasmodium berghei, which makes comparisons difficult. L. major is an intracellular infection of macrophages that causes chronic skin lesions and lymphadenopathy, and there is no lethality during the 12 weeks of experimental infection. P. berghei infection (in BALB/c mice) by contrast is systemic and 100% lethal

by 4 weeks. The difference in the growth of WT and *mif*- *L.major* strains in vivo becomes evident only at 9 weeks after infection (manuscript ref. #35). By contrast, 100% lethality is reached with *PbAwt* and *PbAmif*- at the same 4 week time in the more rapidly progressive and systemically toxic *P. berghei* infection (manuscript ref# 14, please refer to Fig. S3c). As the publication describing the infection by the *mif*- *L. major* strain gives evidence for less exhaustion and depletion of *L. major*-specific CD4+ T cells, we offer only that there is a similarity in mechanism of action of the two examples of parasite-encoded MIF (e.g., PMIF or LmMIF).

The role of PMIF in modulating host immunity in the *P berghei* model is obscured by acute lethality in the first infection, and is not revealed except by antibiotic cure and re-infection (or by adaptive T cell transfer into naïve mice), where there is evidence for a robust recall response (Figs. 3,6,7,8). Severe malaria infection impairs germinal center responses, which is associated with a strong Th1 response that promotes CD4 T cell differentiation into a terminal effector phenotype (manuscript refs. #6-8), with reduced Tfh cell differentiation and B cell antibody response. Our data with *PbAmif*- parasite infection and with PMIF immunization support enhanced germinal centers, and Tfh and B responses as mediating protection from a second infection.

2. The authors test for the role of PMIF in *PbAwt* infection, by trying to neutralize PMIF through immunization of the host with a self-amplifying PMIF gene encoded RNA. The idea is that anti-PMIF antibodies generated here would neutralize PMIF made by the blood-stage plasmodia. This is probably because PMIF is secreted/ surface expressed by blood stage plasmodia (although the protein does not have a known signal peptide / transmembrane domain as per the protein/gene annotation, PBANKA_1444000 on www.PlasmoDB.org). The authors immunized the mice and show that pMIF specific IgG and CD4 T cells responses were elicited (Fig S3a-c). And these mice when challenged with *PbAwt*, but not *Pbmif*- parasites showed better control of the infection (FigS3h). They interpret this as a result of the anti-PMIF IgG generated by immunization, neutralizing the PMIF made by *PbAwt*. However, generating antibodies against a blood-stage parasite protein (PMIF here), leading to the control of the infection is merely the rediscovery of the concept of immunization. The anti-PMIF antibodies may be killing the parasite by targeting its PMIF, similar to what immunizing against any other surface expressed protein like TRAP or PbT may achieve. The authors do not demonstrate actual neutralization of PMIF, nor differentiates the process from a direct killing of the parasite. This alternate interpretation gains more impetus in the context of a similar control of *PbAwt* and *PbAmif*- parasites in a primary infection (as mentioned in (1), above) and the significant boost in anti-PMIF IgG observed in *PbAwt* challenge of PMIF immunized mice (Figure S3B). Given this data set is central to the overall findings of the paper, I have my doubts on the general interpretations of the study.

As in our response to Reviewer #2 (point 2), previously published data on PMIF function using recombinant protein and PMIF gene knockout P. berghei indicate that PMIF is secreted into the infected erythrocyte and released upon schizont rupture (manuscript ref. #13). We performed experiments to further verify surface non-expression of PMIF (see Western blotting data provided for point 2 in response to Reviewer #2). Collectively, these data argue against a neutralization or clearance action of anti-PMIF IgG in eliminating PbAWT-infected RBCs.

*To provide further evidence that anti-PMIF IgG neutralizes PMIF, we show in revision new **Suppl. Fig. 4f**. Anti-PMIF IgG from immune serum inhibits TLR4 induction by recombinant PMIF or PMIF expressed in a PbAWT-infected RBC lysates.*

Malaria itself is known to cause severe immunosuppression in mice. Hence, immunizing against and controlling the infection better is expected to improve surrogate determinants of stronger immune responses. The authors have failed to distinguish this possibility from the proposed neutralization of PMIF to directly improve host immunity.

Though I feel that its flaws do not completely discredit the suggestion of the manuscript that PMIF may be a direct immunomodulator as indicated in figure 1, the somewhat non-critical evaluations of the results that followed might disqualify the mechanism attributed to it.

*We agree with the Reviewer that malaria causes immunosuppression, and we have revised our Discussion to more clearly distinguish the effect of the inhibition of PMIF activity from a classic “immunizing against and controlling the infection”. We stress that prior studies of PMIF (manuscript ref. #14) and PbAmif- parasites support a specific mechanism that interferes with the generation of effective cellular and humoral responses. We submit that the present study, now augmented with additional data, supports a plausible model whereby interference (or absence) of PMIF activity during infection promotes a stronger recall response (cellular and humoral) by increasing Tfh and memory T cell responses. In revision, we provide a scheme to better illustrate the proposed mode of action of PMIF and the impact of its neutralization (**new Suppl Fig. 11**).*

In closing, we trust that we have correctly interpreted the Reviewers’ critique and that we now have a more clear and complete manuscript for consideration. Please do not hesitate to contact us with any questions or for further clarification.

Reviewers' comments:

Reviewer #2 (Remarks to the Author):

The authors have addressed my concerns.

Reviewer #3 (Remarks to the Author):

1. We submit that the tempo and pathology of *Leishmania major* infection differ from that of *Plasmodium berghei*, which makes comparisons difficult. *L. major* is an intracellular infection of macrophages that causes chronic skin lesions and lymphadenopathy, and there is no lethality during the 12 weeks of experimental infection. *P. berghei* infection (in BALB/c mice) by contrast is systemic and 100% lethal by 4 weeks. The difference in the growth of WT and *mif*⁻ *L. major* strains in vivo becomes evident only at 9 weeks after infection (manuscript ref. #35). By contrast, 100% lethality is reached with *PbAwt* and *PbAmif*⁻ at the same 4 week time in the more rapidly progressive and systemically toxic *P. berghei* infection (manuscript ref# 14, please refer to Fig. S3c). As the publication describing the infection by the *mif*⁻ *L. major* strain gives evidence for less exhaustion and depletion of *L. major*-specific CD4⁺ T cells, we offer only that there is a similarity in mechanism of action of the two examples of parasite-encoded MIF (e.g., PMIF or LmMIF). The role of PMIF in modulating host immunity in the *P. berghei* model is obscured by acute lethality in the first infection, and is not revealed except by antibiotic cure and re-infection (or by adaptive T cell transfer into naïve mice), where there is evidence for a robust recall response (Figs. 3,6,7,8). Severe malaria infection impairs germinal center responses, which is associated with a strong Th1 response that promotes CD4 T cell differentiation into a terminal effector phenotype (manuscript refs. #6-8), with reduced Tfh cell differentiation and B cell antibody response. Our data with *PbAmif*⁻ parasite infection and with PMIF immunization support enhanced germinal centers, and Tfh and B responses as mediating protection from a second infection.

Response: Given the data and the view of the authors that PMIF neutralization confers perceivable protection to only reinfections, I feel that the title and intent of the manuscript should be modified to be true to the data presented: 'Neutralization of the Plasmodium-encoded MIF Ortholog Confers Protective Immunity against Malaria Reinfection' would be my suggestion.

2. As in our response to Reviewer #2 (point 2), previously published data on PMIF function using recombinant protein and PMIF gene knockout *P. berghei* indicate that PMIF is secreted into the infected erythrocyte and released upon schizont rupture (manuscript ref. #13). We performed experiments to further verify surface non-expression of PMIF (see Western blotting data provided for point 2 in response to Reviewer #2). Collectively, these data argue against a neutralization or clearance action of anti-PMIF IgG in eliminating *PbAWT*-infected RBCs. To provide further evidence that anti-PMIF IgG neutralizes PMIF, we show in revision new Suppl. Fig. 4f. Anti-PMIF IgG from immune serum inhibits TLR4 induction by recombinant PMIF or PMIF expressed in a *PbAWT*-infected RBC lysates.

Response: I presume the surface mentioned above is the infected RBC surface. *PbMIF* could be expressed on the surface of merozoites, as suggested, or rather not disproved by Fig 4 in ref#13. Anti MSP1-9 antibodies are directly protective, and yet as the authors show (and as has been known), MSP1-19 is not expressed on RBC surface. Given how central a direct neutralization (or not) of *PbMIF* by anti-pMIF antibody generated by immunization is, to the claims of this paper, I feel this needs to be addressed. A western blot similar to the one presented in response to Reviewer 2's concern in the same lines would help, if added to the manuscript, and performed by co-incubating anti-pMIF and anti-MSP-1(as control) antibodies with purified merozoites.

3. Lines 159-179 (*PMIF* Influences *PbA* Development in Liver).

Response: The authors infect mice with PbMIF- parasites and claim that PMIF reduces immunity to liver-stage malaria, with a reduced development of liver-resident CD8 T cells. This seems to be another instance of misinterpretation of the data. No data is presented to help conclude that both PbAwt and PbAmif- parasites have similar (or not) capacity to develop in hepatocytes. It is hard to attribute the delayed patency shown by PbAmif- parasites to a differential CD8 T cell response that is generated. Because this would mean that pmif sufficient parasites (PbAwt) would hinder CD8 T cell responses that would control a primary liver infection. 2 days is the normal length of a liver stage infection in mouse malaria and is too short to make primary CD8 T cell responses relevant. The data presented in Supp fig 3 and the title of the section seems to attribute the difference in PbAmif- parasite development in the liver to CD8 T cell responses. Tackling this question might require the use of radiation attenuated parasites or may have to limit the question to reinfections only.

1) The Referee's concern is stated as:

"I presume the surface mentioned above is the infected RBC surface. PbMIF could be expressed on the surface of merozoites, as suggested, or rather not disproved by Fig 4 in ref #13. Anti MSP1-9 antibodies are directly protective, and yet as the authors show (and as has been known), MSP1-19 is not expressed on RBC surface. Given how central a direct neutralization (or not) of PbMIF by anti-pMIF antibody generated by immunization is, to the claims of this paper, I feel this needs to be addressed. A western blot similar to the one presented in response to Reviewer 2's concern in the same lines would help, if added to the manuscript, and performed by co-incubating anti-pMIF and anti-MSP-1 (as control) antibodies with purified merozoites.

We agree that additional data supporting the presence of PMIF on the merozoite surface will help demonstrate that anti-PMIF antibodies target only PMIF released after schizont rupture. As shown below (and included in new Suppl. Fig 4i), we tested for the presence of PMIF by separating proteins from cytosolic and membrane fractions of purified merozoites by gel electrophoresis and probing with anti-PMIF antibodies. We used MSP-1 as a control for membrane protein. PMIF is detectable only in the cytosolic fraction, and MSP-1 only in the membrane fraction.

2). *The authors infect mice with PbMIF- parasites and claim that PMIF reduces immunity to liver-stage malaria, with a reduced development of liver-resident CD8 T cells. This seems to be another instance of misinterpretation of the data. No data is presented to help conclude that both PbAwt and PbAmif- parasites have similar (or not) capacity to develop in hepatocytes. It is hard to attribute the delayed patency shown by PbAmif- parasites to a differential CD8 T cell response that is generated. Because this would mean that pmif sufficient parasites (PbAwt) would hinder CD8 T cell responses that would control a primary liver infection. 2 days is the normal length of a liver stage infection in mouse malaria and is too short to make primary CD8 T cell responses relevant. The data presented in Supp fig 3 and the title of the section seems to attribute the difference in PbAmif- parasite development in the liver to CD8 T cell responses. Tackling this question might require the use of radiation attenuated parasites or may have to limit the question to reinfections only.*

Ref #15 (cited in *Introduction*, p. 4 and *Results*, p. 7) indicates that PMIF has a role in liver stage parasite development and this is consistent with our observed results (Suppl. Fig 3a). Our intended conclusion from Suppl. Fig 3 is that PMIF absence is associated with an improved CD8 T cell response against *Plasmodium* liver stages (which we measure at 7 days). We fully agree that this CD8 T cell response is not the *cause* of protection, however an augmented CD8 response may be consequential to *second* infection by sporozoites. We regret our mis- or

overstatement. In revision, we temper our interpretation and conclude more simply (see p. 6-7, ***PMIF Influences PbA Development in Liver***) with “These data support the notion that PMIF deficiency impairs liver-stage parasite development, as previously suggested¹⁵, and this impairment is associated with an augmentation in the liver resident CD8 T cell responses”. We hope that this may be acceptable to the Referee and the Editor.

As a point of interest, our ongoing (unpublished) studies indicate that hepatocytes infected with *PbAmif*- parasites undergo increased p53-dependent apoptosis. This is notable since a known action of host (mammalian) MIF is to inhibit p53 (Mitchell *et al.* *PNAS* 99, 345-50). Presumably, *Plasmodium* proteins from apoptotic hepatocytes become more easily presented to resident hepatic CD8+ T cells.

We hope that we adequately addressed the remaining points and stand ready to provide more clarification if needed.